# From Feature Interaction to Feature Generation:
# A Generative Paradigm of CTR Prediction Models

**Mingjia Yin** [1]  **Junwei Pan** [2]  **Hao Wang** [1]  **Ximei Wang** [2]  **Shangyu Zhang** [2]
**Jie Jiang** [2]  **Defu Lian** [1]  **Enhong Chen** [1]

## Abstract

Click-Through Rate (CTR) prediction, a core task in recommendation systems, aims to estimate the probability of users clicking on items. Existing models predominantly follow a discriminative paradigm, which relies heavily on explicit interactions between raw ID embeddings. However, this paradigm inherently renders them susceptible to two critical issues: embedding dimensional collapse and information redundancy, stemming from the over-reliance on feature interactions *over raw ID embeddings*. To address these limitations, we propose a novel *Supervised Feature Generation (SFG)* framework, *shifting the paradigm from discriminative "feature interaction" to generative "feature generation"*. Specifically, SFG comprises two key components: an *Encoder* that constructs hidden embeddings for each feature, and a *Decoder* tasked with regenerating the feature embeddings of all features from these hidden representations. Unlike existing generative approaches that adopt self-supervised losses, we introduce a supervised loss to utilize the supervised signal, *i.e.*, click or not, in the CTR prediction task. This framework exhibits strong generalizability: it can be seamlessly integrated with most existing CTR models, reformulating them under the generative paradigm. Extensive experiments demonstrate that SFG consistently mitigates embedding collapse and reduces information redundancy, while yielding substantial performance gains across various datasets and base models. The code is available at https://github.com/USTC-StarTeam/GE4Rec.

[1] State Key Laboratory of Cognitive Intelligence, University of Science and Technology of China, Hefei, China [2] Tencent Inc, China. Correspondence to: Hao Wang <wanghao3@ustc.edu.cn>.

*Proceedings of the 42nd International Conference on Machine Learning*, Vancouver, Canada. PMLR 267, 2025. Copyright 2025 by the author(s).

## 1. Introduction

Click-Through Rate (CTR) prediction models estimate the probability of users clicking on items based on feature interactions. While conventional wisdom holds that CTR models inherently follow a discriminative paradigm, they are prone to embedding dimensional collapse (Guo et al., 2024) and information redundancy (Zbontar et al., 2021) issues, primarily due to the interactions between raw ID embeddings (Guo et al., 2024).

Different from sequential recommendation models (Kang & McAuley, 2018; Zhai et al., 2024; Yin et al., 2024), very limited research has focused on formulating CTR models under a generative paradigm. This is possibly due to the fact that there are no explicit partial orders among the inputs of CTR models, making it difficult to directly fit them into the popular next-token prediction framework (Vaswani et al., 2017; Kang & McAuley, 2018).

Next-token prediction is an autoregressive generative paradigm (Fig. 1(a)) that treats the sequence up to position $N$ as the source $x_{\text{source}}$ and the token at position $N+1$ as the target $x_{\text{target}}$, using the former to predict the latter. Recently, computer-vision researchers have reconsidered the ordering of image data to better align this modality with the autoregressive framework. For instance, MAR (Li et al., 2024b) (Fig. 1(b)) employs unmasked image tokens as $x_{\text{source}}$ to generate masked tokens as $x_{\text{target}}$, whereas VAR (Tian et al., 2024) (Fig. 1(c)) introduces a next-scale prediction framework that establishes a "coarse-to-fine" ordering. These works design generative frameworks that align with inherent data characteristics, rather than rigidly imposing the next-token prediction paradigm.

This inspired us to reconsider the "order" or "inherent structure" of CTR prediction data. Specifically, CTR prediction handles the multi-field categorical data (Zhang et al., 2016; Pan et al., 2018), where there are usually no explicit partial orders between input features. However, there are complex co-occurrence relationships among those features (Rendle, 2010), which motivates us to treat one side of the co-occurrence features as $x_{\text{source}}$ and the other side as $x_{\text{target}}$. Such a method can be regarded as a "feature generation"

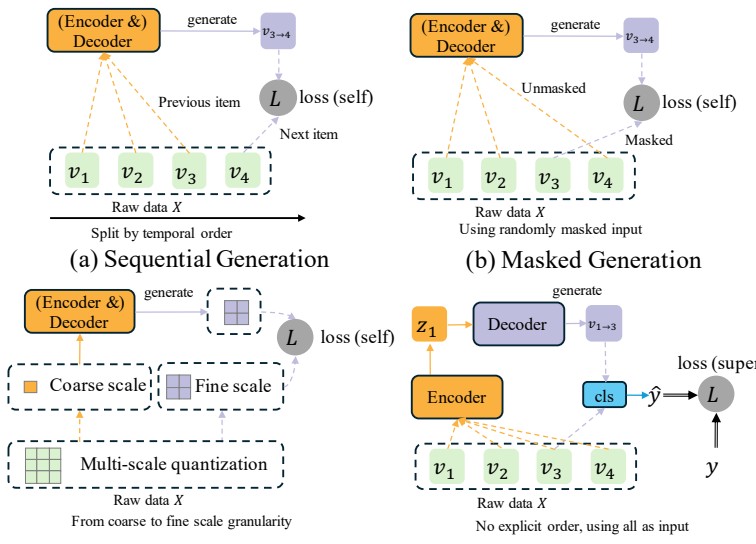

(a) Sequential Generation

(b) Masked Generation

(c) Coarse-to-fine Generation

(d) Feature Generation

*Figure 1.* **Different generative paradigms**. Specifically, this framework employs an (optional) *encoder* to process *source* data of a particular form and generate an output embedding. Subsequently, a *decoder* utilizes this embedding to generate the *target*, representing data of a different form. Finally, a loss function will be used to evaluate the generation quality. Lacking an explicit data structure, our feature generation paradigm adopts an "all predict all" paradigm to predict each feature with all features. Notably, we employ a supervised generative loss function, optimizing the cross-entropy loss regarding the sample-wise label $y_{sup}$, rather than the self-supervised loss.

paradigm, which models the data distribution $P(\mathcal{X})$ on the unordered multi-field categorical data.

Specifically, the feature-generation paradigm employs an *Encoder* to transform $x_{source}$ into a latent space, yielding a new representation for each feature. A *Decoder* then projects these latent representations back to the original space to *generate* all original features as $x_{target}$. Since the *Encoder* takes all original features as input and simultaneously generates all original features, this framework adheres to an *All-Predict-All* framework. Such a design *avoids direct interaction (product) between vanilla ID embeddings* as done in traditional CTR models (Rendle, 2010; Guo et al., 2017; Sun et al., 2021; Wang et al., 2021), and hence prevents the embeddings from dimensional collapse due to Interaction-Collapse Theory (Guo et al., 2024). Moreover, the *Encoder* produces sample-specific representations that are decorrelated from the original embeddings, reducing the redundancy of feature embeddings.

Conventionally, generative paradigms are accompanied by a self-supervised loss. In the next-prediction paradigm in sequential modeling, the label of the next item is usually a self-supervised signal, that is, whether the next item is the ground-truth one or just a random one. However, in CTR prediction, it is unnecessary since natural supervised signals, *i.e.*, click or not in CTR prediction, already exist. Therefore, we adopt a supervised loss with the proposed feature generation paradigm. The feature generation, together with the supervised loss, leads to a novel *Supervised Feature Generation* framework for CTR prediction.

This framework can reformulate nearly every existing feature interaction model, ranging from FM to DeepFM, xDeepFM, and DCN V2. Comprehensive experiments demonstrate that this new framework significantly improves performance, achieving an average of 0.272% AUC lift and 0.435% Logloss reduction, while incurring only a marginal increase in computational overhead—an average increase of 3.14% in computation time and 1.45% in GPU memory consumption. It can produce feature embeddings with reduced collapse and redundancy compared to raw ID embeddings. Additionally, we conduct extensive ablation studies to validate the framework design. We successfully deployed it to Tencent's advertising platforms for click prediction, with a 2.68% GMV lift on a primary scenario, leading to one of the largest revenue lift in 2024.

## 2. Preliminaries

In this section, we present the traditional CTR prediction problem in the discriminative paradigm and its challenges.

### 2.1. Problem definition

The CTR prediction task predicts the probability of users clicking items based on multiple features. It can be formally defined using features $\mathcal{X} \in \{0, 1\}^N$ and a label set $\mathcal{Y} \in \{0, 1\}$, indicating whether users click the candidate item. Typically, $\mathcal{X}$ consists of hundreds of features from user, item, or the context sides, each belonging to a feature field.

### 2.2. CTR prediction in a discriminative paradigm

**Formulation.** Existing CTR models are usually formulated under a discriminative form, by first conducting explicit and/or implicit feature interactions between all features through $g_{inter}(\cdot)$, then employing a classifier $f_{cls}$, *e.g.*, MLPs, upon its output, to get the final prediction score, and lastly optimizing a binary cross-entropy loss regarding the

supervised label $y_{\text{sup}}$:

$$\mathcal{L}_{\text{BCE}}(y_{\text{sup}}, f_{\text{cls}}(g_{\text{inter}}(\{\boldsymbol{v}_i\}))). \tag{1}$$

where $\boldsymbol{v}_i \in \mathbb{R}^d$ is the embedding for feature $i, d$ denotes the embedding dimension. Taking the classic DCN V2 (Wang et al., 2021) model as an example, it can be formalized as:

$$\mathcal{L}(y_{\text{sup}}, \text{DNN}(\sum_{l=1}^{L}\sum_{i=1}^{N}\sum_{j=1}^{N} \boldsymbol{v}_j \odot \boldsymbol{v}_i^{(l)} \boldsymbol{M}_{F(i) \to F(j)}^{(l)})), \tag{2}$$

where $L$ denotes the number of cross layers; $l$ denotes the layer index; $N$ denotes the total number of features; $i$ and $j$ denote the feature indices; $\mathbf{v}_j^{(l)}$ denotes the embedding of the $j$-th feature in the $l$-th layer; $M_{F(i) \to F(j)}^{(l)}$ denotes the projection matrix between the $F(i)$ and $F(j)$ field pair in the $l$-th layer; and $F(i)$ and $F(j)$ denote the field of the feature $i$ and $j$, respectively.

**Limitations of the Discriminative Paradigm.** Despite the great success of many existing discriminative models for CTR prediction (Rendle, 2010; Juan et al., 2016; Pan et al., 2018; He & Chua, 2017; Guo et al., 2017; Lian et al., 2018; Sun et al., 2021; Wang et al., 2021), they still suffer from the following issues:

1. **Dimensional Collapse due to raw ID embedding Interaction**. Existing CTR models usually capture the correlation between features via the feature interaction function. The feature embeddings in these models tend to span a low-dimensional space due to the Interaction-Collapse-Theory (Guo et al., 2024): feature fields with low information abundance, *i.e.*, with collapsed dimensions, constrain the information abundance of other fields.

2. **Limitation to learn data distribution**. Discriminative paradigms learn the distribution $P(\mathcal{Y} \mid \mathcal{X})$ while ignoring $P(\mathcal{X})$, focusing solely on establishing a decision boundary for classification (Harshvardhan et al., 2020; Oussidi & Elhassouny, 2018). This makes these models unable to capture the rich co-occurence correlation between the input (features), and hence limits the quality of the learned representations.

3. **Information redundancy**. Redundancy-reduction principle (Barlow et al., 1961) has been fruitful in different application domains (Barlow, 2001; Grill et al., 2020; Zbontar et al., 2021). This principle necessitates minimizing information redundancy between the two views, that is, their mutual correlation. We have empirically verified this principle in Sec. 4.3.2, and find that models with redundancy-reduced interacted embeddings achieve better recommendation performance (Fig. 4). Existing models mainly use the same raw ID embeddings in the interaction function, exhibiting a strong tendency towards learning redundant representations.

These limitations call for rethinking existing feature interaction models in CTR prediction and motivate us to resort to generative approaches.

## 3. Method

### 3.1. Revisit Generative Paradigm for Recommendation

Next-item prediction is one of the most popular generative paradigms, which predicts the next item in a sequence based on preceding inputs, and is widely adopted in recent generative recommendation models (Zhai et al., 2024; Rajput et al., 2023). Recently, VAR (Tian et al., 2024) reconsidered the "order" of images and proposed a next-scale prediction paradigm. It designs generative frameworks that align with inherent data characteristics, *i.e.*, the "multi-scale, coarse-to-fine nature" of images. As long as the data order $x_{\text{source}} \prec x_{\text{target}}$ is defined, a generative model can be applied to generate $x_{\text{target}}$ upon $x_{\text{source}}$.

This inspired us to reconsider the "order" or the "inherent data structure" in CTR prediction. The data used in CTR prediction is the multi-field categorical data (Zhang et al., 2016; Pan et al., 2018), with categorical features from the user, item, and context sides. Even though there are hierarchies between some features, for example, the `item ID` belongs to a `fine-grained category ID`, and the latter belongs to a `coarse-grained category ID`, there is no global partial order between all features.

However, the most essential characteristic of multi-field categorical data is the co-occurrence of features (Rendle, 2010; Koren et al., 2009). Denote the active features of the $s$-th sample (i.e., an explicit click or an implicit non-click) as $X^{(s)} = \{x_i\}$, the joint occurrence of each feature pair, such as $(x_i, x_j)\colon x_i, x_j \in X^{(s)}$ denotes the collaborative signal regarding users' explicit or implicit feedbacks. Such co-occurrence of features is the inherent "order", or more precisely, the inherent "structure" of data in CTR prediction.

### 3.2. The Feature-Generation Paradigm

To this end, we propose treating one side of the co-occurrence features as the *Source* and the other side as the *Target*. Specifically, given all feature pairs $\{(x_i, x_j)|\forall(i,j)\}$ and their corresponding embedding $\{(\boldsymbol{v}_i, \boldsymbol{v}_j)|\forall(i,j)\}$, we define $x_{\text{source}}$ and $x_{\text{target}}$ as :

$$x_{\text{source}} = \{\boldsymbol{v}_i \mid i \in \{1, 2, \ldots, N\}\}, \tag{4}$$

$$x_{\text{target}} = \{\boldsymbol{v}_j \mid j \in \{1, 2, \ldots, N\}\}, \tag{5}$$

where $N$ is the number of features, and $\boldsymbol{v}_i$ and $\boldsymbol{v}_j$ is the embedding vector of feature $i$ and $j$.

Given $x_{\text{source}}$ and $x_{\text{target}}$, we employ an *Encoder* $f_{\text{encoder}}\colon \mathcal{V} \to \mathcal{Z}$ to transform $x_{\text{source}}$ into a hidden space

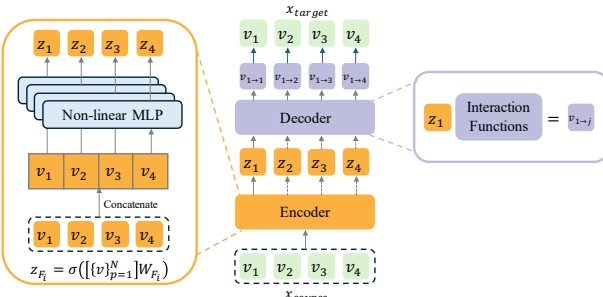

Figure 2. The feature generation framework builds an *encoder* based on all features as the $x_{\text{source}}$, generates an output embedding, and utilizes it to predict all features simultaneously as the $x_{\text{target}}$. For multi-layer generation, generated representations in each layer will serve as the $x_{\text{source}}$ and $x_{\text{target}}$ in the next layer generation. Specifically, the *encoder* is implemented as a field-wise single-layer non-linear MLP, while the *decoder* is implemented with feature interaction functions in previous CTR models.

$\mathcal{Z}$ to build new representations for each feature. We then employ a *Decoder* $f_{\text{decoder}} \colon \mathcal{Z} \to \mathcal{V}$ to map the hidden space $Z$ back to the original input space $\mathcal{V}$, to *generate* all features. Taking feature $i$ as an example, the encoder constructs a new representation $z_i$ for $i$, and the decoder further maps $z_i$ to generated representation $f_{\text{decoder}}^{i \to j}(f_{\text{encoder}}(v_i))$, or $v_{i \to j}$ for abbreviation. Notably, each encoded feature representation generates all features, *i.e.*, $\{x_j\}$, simultaneously, which follows an *All Predict All* framework. Implementations of the *Encoder* and *Decoder* will be detailed in Section 3.3.

Finally, a classifier is employed to calculate the relevance (*e.g.*, dot or Hadmard product, or a simple concatenation) between the output of the decoder, *i.e.*, $v_{i \to j}$, and the target representation $v_j$, pooling these results across all feature pairs $\{(i, j)\}$, and then get the final prediction score. The choice of classifier largely depended on the architecture of the original CTR model, and corresponds to the Layer Pooling and Layer Aggregator in (Kang et al., 2025). For example, in the generative FM, we just need to sum up the inner products between $v_{i \to j}$ and $v_j$ across all feature pairs, *i.e.*, $\hat{y} = \sigma(\sum_i \sum_j \langle v_{i \to j}, v_j \rangle)$, where $\sigma$ is the sigmoid function. We ignore the linear and bias terms here for simplicity.

The entire generative paradigm is formalized in Eq. 3 and depicted in Fig. 2. In Eq. 3, we first employ $f_{\text{encoder}_i}$ to transform $x_{\text{source}}$ (the set of all feature embeddings $\{v_k\}_{k=1}^N$) to obtain hidden representation of feature $i$. $f_{\text{decoder}_{i \to j}}$ further maps the hidden representation back to the original input space. Then, a classifier $f_{\text{cls}}$ will be used to calculate the final prediction score. Finally, the entire model will be optimized with a supervised label $y_{\text{sup}}$.

### 3.3. Architecture Design

**Encoder** The encoder transforms the input into a hidden space, generating new representations for each feature. We employ a simple field-wise single-layer non-linear MLP, which takes all feature representations as input, then applies a field-wise projection matrix $W_{F(i)}$ and a ReLU activation function. Specifically, the encoder for feature $i$ is:

$$f_{\text{encoder}_i}([\boldsymbol{v}]) = \sigma([\{\boldsymbol{v}_k\}_{k=1}^N]W_{F(i)}), \tag{6}$$

where $\sigma$ is the ReLU activation function, $[\{v_k\}_{k=1}^N] \in \mathbb{R}^{Nd}$ denotes the concatenation of all feature embeddings[1], $F(i)$ denotes the field of feature $i$, and $W_{F(i)} \in \mathbb{R}^{Nd \times d}$ denotes the projection matrix for the field of feature $i$. The Eq. 6 follows the principle of minimal sufficient complexity: ablation of any constituent element results in significant performance deterioration, whereas augmentations with more sophisticated components fail to demonstrate performance gains. Refer to Sec. 4.4 for details.

**Decoder** The decoder aims to transform the encoder output in a hidden space $\mathcal{Z}$ back to the original space $\mathcal{V}$. It may consist of multiple stacked layers, with each layer conducting the space mapping through a projection matrix $W$:

$$f_{\text{decoder}_{i \to j}}(\boldsymbol{z}_i) = \boldsymbol{z}_i \boldsymbol{W}. \tag{7}$$

Notably, $W$ with different properties correspond to various feature interaction functions(Wang et al., 2021; Kang et al., 2025). For example, a field-pair wise scale identity matrix $W := \text{diag}(w_{F(i) \to F(j)}, \dots, w_{F(i) \to F(j)}) =$

---

[1]For single-value fields, we can simply concatenate the embeddings of the active features from them; while for multi-value fields, we need to first aggregate the multiple values for each field through sum or mean pooling.

**E.g.,** $f_{\textbf{encoder}_i}([\boldsymbol{v}]) = \sigma([\{\boldsymbol{v}_k\}_{k=1}^N]W_{F(i)})$

$x_{\textbf{source}}$  $x_{\textbf{target}}$

**Supervised label**

$$\mathcal{L}(\ y_{\text{sup}}\ ,\ f_{\text{cls}}\ (\ f_{\text{decoder}_{i \to j}}\ (\ f_{\text{encoder}_i}\ (\ \{\boldsymbol{v}_k\}_{k=1}^N\ )),\ \boldsymbol{v}_j\ )) \tag{3}$$

**Classifier**

**E.g.,** $f_{\textbf{decoder}_{i \to j}} = W_{F(i) \to F(j)} \in \mathbb{R}^{K \times K}$

$w_{F(i) \to F(j)} \mathcal{I}$ correspond to the field-weighted interaction function used in FwFM (Pan et al., 2018) and xDeepFM (Lian et al., 2018), while a field-pair wise full matrix $\boldsymbol{W} := \boldsymbol{W}_{F(i) \to F(j)}$ correspond to the one used in FmFM (Sun et al., 2021) and DCN V2 (Wang et al., 2021). Using FmFM as an example, it can be reformulated into our feature generation paradigm as:

$$\mathcal{L} \left( y, \sum_{i,j=1}^{N} \left( \sigma([\{\boldsymbol{v}_k\}_{k=1}^{N}] W_{F(i)}) W_{F(i) \to F(j)} \right) \odot \boldsymbol{v}_j \right). \tag{8}$$

More formal generative reformulations of popular CTR models can be found in Appendix A.

### 3.4. Loss

Existing generative approaches for recommendation usually employ self-supervised losses (Kang & McAuley, 2018; Zhai et al., 2024). For example, next-item prediction treats the next item as an unsupervised label and adopts a cross-entropy loss to evaluate the generation quality. However, in our feature generation paradigm, we need to be careful when we try to employ such a self-supervised loss. For example, if we follow the next-item prediction, and use a similar "another feature prediction" loss, that is, utilize the encoder to predict whether the another feature is true or not, then *it would lead to label leakage since the true "another feature" is already in the input*. Instead, employing a supervised loss in our paradigm would force the encoder to learn the collaborative information about the input regarding the supervised label $y_{\text{sup}}$.

### 3.5. Discussion

**Representation Learning** Our core contribution is to employ an Encoder network to build a new representation for each feature, thereby overcoming the limitation of original ID embeddings (Guo et al., 2024). Many existing works also pay attention to constructing new representations. Guo et al. (2024) proposed to build several independent embeddings $\{\boldsymbol{v}_i^{(m)}\}$ for each feature $i$ by constructing multiple independent embedding tables, and such multi-embedding paradigm can be further attributed back to FFM (Juan et al., 2016). Rajput et al. (2023) proposed to first map the LLM embedding of each feature into a hidden space through an RQ-VAE, and then quantize the hidden representation to discrete Semantic IDs. We study the quality of these newly built embeddings by the dimensional collapse analysis (Sec. 4.3.1) and redundancy-reduction analysis (Sec. 4.3.2).

**Relationship with Gating Mechanism** Our encoder can be seen as a generalization to existing works (Huang et al., 2019; Mao et al., 2023; Chang et al., 2023), which treat it as a gating mechanism to generate attentive weights. For example, Fibinet (Huang et al., 2019) introduced a SENET mechanism, which also resembles our encoder, to "pay more attention to the feature importance". Inspired by LHUC (Swietojanski et al., 2016), PEPNet (Chang et al., 2023) proposed a Gate Neural Unit to "personalize network parameters", which takes the domain-side features as inputs, conducts nonlinear activation, and then interacts with the results with the embeddings of the backbone models. Mao et al. (2023) employed a context-aware feature aging layer for feature selection. Our work offers an alternative representation learning interpretation of these methods.

## 4. Experiments

In this section, we aim to address these research questions:

- **RQ1**: To what extent can the paradigm shift improve existing discriminative feature interaction models?

- **RQ2**: Can the generative paradigm mitigate the inherent drawbacks of raw ID embeddings in discriminative paradigms, specifically in terms of embedding dimensional collapse and information redundancy reduction?

- **RQ3**: Is the current paradigm design optimal for feature generation? What will happen if we use different $x_{\text{source}}$, *Encoder*, or $x_{\text{target}}$?

### 4.1. Setup

**Datasets & Evaluation protocols.** In this work, we have conducted experiments based on two widely adopted large-scale datasets, namely Criteo (cri, 2014) and Avazu (ava, 2014). Dataset statistics are summarized in Appendix B.1. As for evaluation, we evaluate the recommendation performance with AUC and Logloss.

**Baselines.** To verify versatility of our method, we integrate it with various representative models, including explicit feature interaction models FM (Rendle, 2010), FmFM (Sun et al., 2021), CrossNetv2 (Wang et al., 2021), and DNN-based models DeepFM (Guo et al., 2017), IPNN (Qu et al., 2016), xDeepFM (Lian et al., 2018), DCN V2 (Wang et al., 2021). All experiments are based on a popular library FuxiCTR (Zhu et al., 2020; 2022). More details can be found in Appendix B.2. Besides, the computational complexity are provided in Sec. B.4.

### 4.2. Recommendation performance comparison between discriminative and generative paradigms (RQ1)

**Offline results.** We apply the feature generation framework with various recommendation models, with results

*Table 1.* Recommendation performance of models with the DIScriminative (DIS) and GENrative (GEN) paradigm. We conduct a two-tailed T-test to calculate the statistical significance, with results presented in a form of *mean(variance)*. **Bolded values** refer to the best performance, and * means the corresponding p-values are less than 0.05.

| | Model | | Criteo | | Avazu | |
| --- | --- | --- | --- | --- | --- | --- |
| | | | AUC↑ | Logloss↓ | AUC↑ | Logloss↓ |
| Explicit | FM | DIS | 0.80236(9e-05) | 0.44889(7e-05) | 0.78877(1e-04) | 0.37529(4e-05) |
| | | GEN | **0.81108(1e-04)*** | **0.44077(1e-04)*** | **0.79260(1e-04)*** | **0.37279(6e-05)*** |
| | FmFM | DIS | 0.80552(3e-04) | 0.44626(3e-04) | 0.78990(2e-04) | 0.37519(7e-04) |
| | | GEN | **0.80992(7e-04)*** | **0.44258(8e-04)*** | **0.79266(9e-05)*** | **0.37287(2e-04)*** |
| | CrossNet V2 | DIS | 0.81312(1e-04) | 0.43918(2e-04) | 0.79106(1e-04) | 0.37319(2e-04) |
| | | GEN | **0.81540(4e-05)*** | **0.43661(5e-05)*** | **0.79301(2e-04)*** | **0.37200(5e-05)*** |
| DNN-based | DeepFM | DIS | 0.81380(8e-05) | 0.43804(6e-05) | 0.79285(1e-04) | 0.37224(1e-04) |
| | | GEN | **0.81396(6e-05)*** | **0.43788(5e-05)*** | **0.79333(7e-05)*** | **0.37181(1e-04)*** |
| | xDeepFM | DIS | 0.81365(1e-04) | 0.43819(1e-04) | 0.79222(1e-04) | 0.37246(5e-05) |
| | | GEN | **0.81421(7e-05)*** | **0.43775(9e-05)*** | **0.79429(1e-04)*** | **0.37123(7e-05)*** |
| | IPNN | DIS | 0.81341(5e-05) | 0.43850(2e-05) | 0.79348(3e-04) | 0.37159(1e-04) |
| | | GEN | **0.81415(8e-05)*** | **0.43776(1e-04)*** | **0.79451(8e-05)*** | **0.37105(1e-04)*** |
| | DCN V2 | DIS | 0.81387(6e-05) | 0.43826(4e-05) | 0.79282(2e-04) | 0.37222(1e-04) |
| | | GEN | **0.81472(6e-05)*** | **0.43713(5e-05)*** | **0.79342(5e-05)*** | **0.37180(5e-05)*** |

presented in Tab.4. Overall, the proposed method exhibits promising effectiveness and achieves consistent performance lift across different models, achieving an average of 0.272% AUC lift and 0.435% Logloss reduction. Usually, a 0.1% AUC (gAUC) lift is regarded as a huge improvement in recommendation systems (Zhu et al., 2022).

Specifically, the generative paradigm on explicit feature interaction models can bring an average of 0.428% AUC lift and 0.689% Logloss reduction. Notably, DCN V2 incorporates a DNN based on CrossNet, enhancing its modeling capability. The discriminative DCN V2 surpasses the discriminative CrossNet by 0.157% lift in AUC and 0.235% in Logloss reduction. Surprisingly, when CrossNet is reformulated within our generative paradigm, it can even outperform the discriminative DCNv2 by 0.106% lift in AUC and 0.089% reduction in Logloss, verifying the promising potential of generative paradigms. For DNN-based models, the improvement is less pronounced. Nevertheless, even with complex DNN-based models, the paradigm shift still brings significant enhancements, achieving an average improvement of 0.116% in AUC and 0.181% in Logloss reduction.

Moreover, our generative paradigm can narrow the performance gap caused by different model architectures. For example, the strongest discriminative model (DCNv2) surpasses the weakest (FM) by 1.151% AUC on Criteo, whereas the gap between their generative counterparts shrinks to 0.364%. In detail, Appendix B.3 compares the

coefficient of variation for each paradigm, and Fig. 6 shows that the results of the generative paradigm are markedly smaller, corroborating its ability to reduce the gap between different models. These findings further underscore the importance of the paradigm shift.

In summary, our framework consistently improves the performance of various existing feature interaction models under the original discriminative paradigm and markedly narrows inter-architectural performance gaps.

**Online A/B Testing.** We deployed the proposed generative paradigm in one of the world's largest advertising platforms. The production model employs Heterogeneous Experts with Multi-Embedding architecture (Guo et al., 2024; Su et al., 2024; Pan et al., 2024). We switch the IPNN expert in the production model into a generative paradigm, which models the interactions between more than five hundred user-, ad-, and context-side features. During the one-week 20% A/B testing, demonstrated promising results, achieving 2.68% GMV lift and 2.46% CTR lift on several vital scenarios, including Moments pCTR, Content and Platform pCTR, and DSP pCTR. These improvements were statistically significant according to t-tests. The proposed feature generation framework has been successfully deployed as the production model in the above-mentioned scenarios, leading to a revenue lift by hundreds of millions of dollars per year.

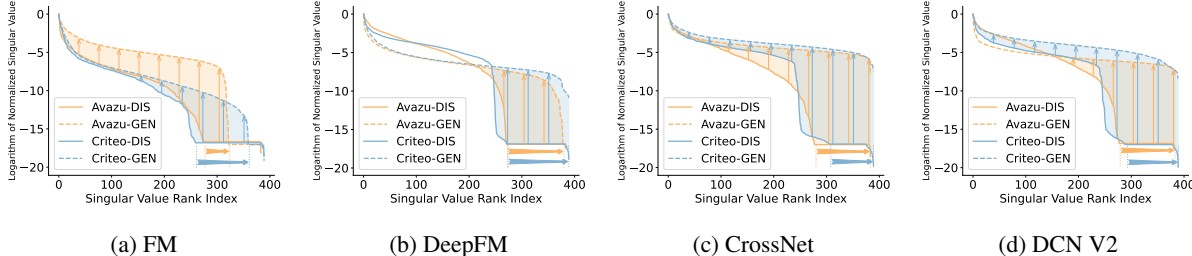

*Figure 3.* Normalized singular value spectrum of embeddings used to interact with raw ID embeddings. It is the concatenation of raw ID embeddings for the discriminative paradigm, while the embedding immediately constructed by the *encoder* for the generative paradigm.

## 4.3. How does the generative paradigm work? (RQ2)

### 4.3.1. GENERATIVE PARADIGM MITIGATES EMBEDDING DIMENSIONAL COLLAPSE

**Dimensional collapse evaluation protocols.** Dimensional collapse means that the embeddings only span a low-dimensional subspace of the available representation space (Jing et al., 2021; Guo et al., 2024). With a slight abuse of notation, we will detail how to measure the dimensional collapse issue by singular value decomposition. Specifically, we evaluate the dimensional collapse issue at the sample level. We begin by obtaining the sample embedding matrix $Z \in \mathbb{R}^{B \times d}$ using the validation dataset, where $B$ denotes the batch size and $d$ the dimension size (Notably, this batch-wise setting will greatly enhance the analysis efficiency, and we have verified the robustness of this setting in Appendix B.5). The covariance matrix is then derived as $C = \frac{1}{B} \sum_{i=1}^{B} (z_i - \bar{z})(z_i - \bar{z})^T$, with $\bar{z} = \frac{1}{B} \sum_{i=1}^{B} z_i$. Subsequently, we determine the singular values $S = \text{diag}(\sigma^k)$ of $C$ via singular value decomposition (SVD) and normalize them by the maximum singular value: $S' = \text{diag}\left(\frac{\sigma^k}{\max(\sigma^k)}\right)$. Finally, we present these normalized singular values in descending order, as shown in Fig. 3.

**Evaluated embeddings.** We focus on the direct impact of the feature generation framework on the embedding space. Specifically, we analyze the embedding used to interact with raw ID embeddings. In the discriminative paradigm, it is the concatenation of raw ID embeddings, formally defined as $[\{v_k\}_{k=1}^{N}]$. In the generative paradigm, it is the concatenation of feature embedding immediately constructed by the *Encoder*, i.e., $[\{z_k\}_{k=1}^{N}]$. We study this embedding to investigate the direct influence of the generative paradigm.

**Generative paradigm mitigates dimensional collapse.** For brevity, we illustrate the singular value spectrum of the embedding space for four representative models in Fig. 3. Visualization of all models can be found in Appendix C.1. In each sub-figure, the spectrum exhibits a rapid decay. Taking Fig. 3d as an example, the singular values of DCN V2 on Criteo remain high up to index 250, with values around

$1 \times 10^{-5}$. However, they drop dramatically to $1 \times 10^{-15}$ at index 280, a reduction of $10^{10}$ times. This indicates an extreme imbalance among dimensions, *i.e.*, only a minority of dimensions dominate the embedding space. After index 280, the singular values remain around $1 \times 10^{-15}$, essentially zero. These singular values account for approximately 30% of the total singular values, implying that 30% of the dimensions in the embedding carry no meaningful information, which is clearly unfeasible.

These phenomena are significantly mitigated in the generative one. With the exception of FM, the singular value spectra of the other methods do not exhibit the abrupt decay mentioned earlier. Instead, they decline at a relatively slower rate, indicating a more balanced embedding space. Even for simple models like FM, our generative paradigm can increase the number of meaningful dimensions by 25%. We attribute this improvement to the integration of all feature fields when constructing embeddings using the feature generation framework. We conclude the following result:

> *Result 2. The generative paradigm substantially mitigates the issue of embedding dimensional collapse.*

### 4.3.2. REDUNDANCY REDUCTION VIA GENERATIVE FEATURE LEARNING

**Information redundancy evaluation protocols.** According to the information redundancy reduction principle (Barlow et al., 1961; Zbontar et al., 2021), the two interacted embeddings are expected to exhibit low correlation. To quantify this, we employ the Pearson Correlation Coefficient between each dimension of the two interacted embeddings, defined as $\rho_{X,Y} = \frac{\text{Cov}(X,Y)}{s_X s_Y}$. For the discriminative paradigm, $X$ and $Y$ are the two interacted embeddings, while $X$ and $Y$ are the transformed $x_{\text{source}}$ and $x_{\text{target}}$ embeddings for the generative paradigm. $s_X$ and $s_Y$ denote their respective standard deviations.

**Negative connection between redundancy metric and recommendation performance.** We have visualized the correlation matrix of FM, DeepFM, DCN V2 in Fig. 4. More visualizations are provided in Appendix C.3. In Fig. 4a,

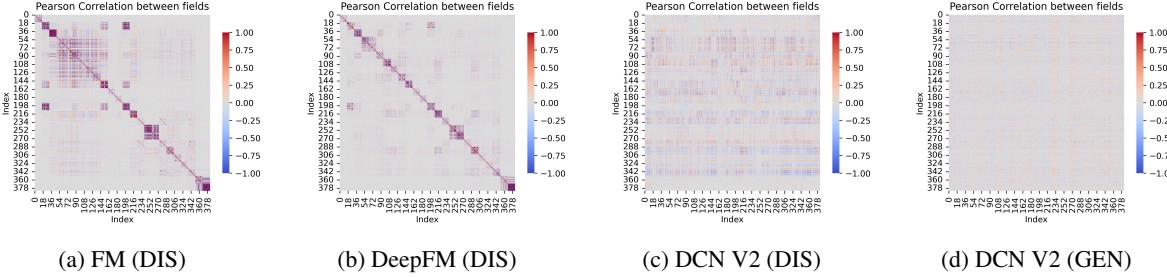

|  |  |  |  |
|---|---|---|---|
| (a) FM (DIS) | (b) DeepFM (DIS) | (c) DCN V2 (DIS) | (d) DCN V2 (GEN) |

*Figure 4.* Pearson correlation matrix between two interacted embeddings. (a)→(b)→(c) means more complex models, which also exhibits a trend of redundancy reduction. This reveals the importance of information redundancy reduction when designing CTR models. In (d), we can find the generative DCN V2 almost produces a zero correlation matrix, perfectly aligning with the redundancy reduction principle.

we have derived two major observations: (1) Intra-field correlation. It forms some obvious diagonal blocks, while each block corresponds to a feature field. This means the information within a feature field is highly correlated, *i.e.*, redundant information. (2) Inter-field correlation. The index 32 - 160 forms a big diagonal block, which is exactly the correlation between field with index 3 - 10. This means these feature fields are also highly correlated, violating the redundancy reduction principle. These observations may explain the inferior performance of FM.

Then we analyze by comparing different models. DeepFM builds a parallel DNN upon FM, which greatly decreases inter-field correlation and increases recommendation performance. DCN V2 further incorporates a more advanced explicit feature interaction module based on DeepFM. In Fig. 4c, the diagonal blocks representing intra-field correlation are almost reduced, which explains the recommendation performance lift. *All these results reveal a negative connection between the redundancy metric and recommendation performance, which can guide model designing.*

**Generative paradigm reduces information redundancy.** Despite the transformations applied to raw ID embeddings in DCN V2, we still observe correlations in Fig. 4c. In contrast, the correlation matrix is nearly a zero matrix in Fig. 4d, indicating that the two vectors are highly de-correlated and thus adhere to the redundancy reduction principle. This demonstrates our framework's ability to reduce information redundancy effectively. We conclude the following result:

> *Result 3. The feature generation framework produces embeddings highly de-correlated with raw ID embeddings, adhering to the redundancy reduction principle.*

### 4.4. Ablation on the feature generation framework design (RQ3)

For simplicity, all ablation studies are based on DCN V2.

**Ablation on the $x_{\text{source}}$ design.** As stated in Discussion 1 of discriminative paradigms, one of their major limitations is the direct interactions between raw ID embeddings, especially those of low-cardinality fields. To tackle this issue, we propose to utilize all field embeddings as $x_{\text{source}}$ to build new representations for all fields. For comparison, we will investigate the following configurations to reveal the significance of our design: (a) using only each field's embedding as $x_{\text{source}}$ for all fields; (b) using all field embeddings as $x_{\text{source}}$ for 10 fields with the highest cardinality; and (c) using all field embeddings as $x_{\text{source}}$ for 10 fields with the lowest cardinality. Results are presented in Fig. 5a.

We can observe that using all field embeddings as $x_{\text{source}}$ outperforms other settings, revealing the necessity of constructing embeddings with all features. Additionally, the results of only constructing low-cardinality fields with all features are significantly better than the high-cardinality counterpart, which corroborates our previous assertion that low-cardinality fields suffer more from severe information insufficiency. We conclude the following result:

> *Result 4. Treating all feature fields as $x_{source}$ is effective for feature generation. In particular, low-cardinality field embeddings suffer from more severe issues than high-cardinality field ones, underscoring the importance of constructing new embeddings for these fields.*

**Ablation on the *Encoder* design.** The adopted *Encoder* is a field-wise one-layer non-linear MLP. To further investigate its properties, we first construct the following model variants: (b.1) using a field-shared MLP, (b.2) removing non-linear activations, and (b.3) stacking one more layer. In Fig. 5b, simplifying the *Encoder* with either (b.1) or (b.2) leads to significant performance degradation. The former underscores the importance of constructing distinct embeddings for different fields, which aligns with our intuition. The latter highlights the necessity of modeling non-linear relationships among features, which also contributes significantly to alleviating the dimensional collapse issue, as

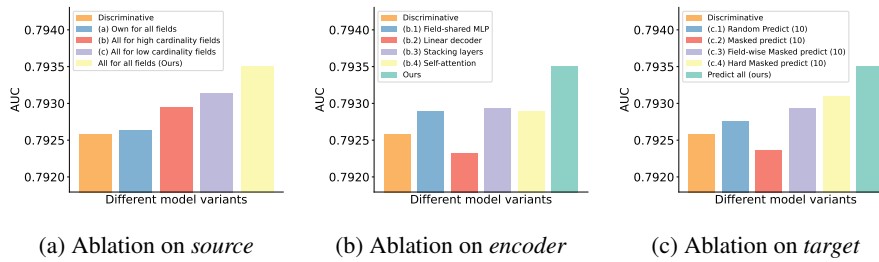

(a) Ablation on *source*      (b) Ablation on *encoder*      (c) Ablation on *target*

*Figure 5.* Ablation study on the feature generation framework design using DCN V2 on Avazu.

verified in Appendix C.4. On the other hand, increasing the complexity of *Encoder* with (b.3) even greatly degrades recommendation performance, AUC decreases from 0.7935 to 0.7929, which may be caused by over-fitting.

Next, we investigate whether other generative models are feasible: (b.4) self-attention networks. In Fig. 5b, we observe that both (b.4) outperform the original discriminative paradigm but underperform the generative paradigm with the MLP-based encoder. This confirms the effectiveness of the generative paradigm and further demonstrates that our encoder is a simple yet effective design for constructing meaningful embeddings. We conclude the following result:

> *Result 5. The field-wise non-linear one-layer MLP is a simple yet effective* Encoder. *Common modifications, including simplification or increased complexity, lead to inferior recommendation performance.*

**Ablation on the $x_{\text{target}}$ design.** In our paradigm, we generate all features simultaneously. We compare different implementations of $x_{\text{target}}$: (c.1) "predict-random-selected": generating only randomly selected feature fields; (c.2) "masked feature modeling": randomly masking some fields in $x_{\text{source}}$ with a learnable mask vector and predicting them as $x_{\text{target}}$, akin to masked image modeling (He et al., 2022); (c.3) "field-aware masked feature modeling": similar to (c.2) but using field-specific mask vectors; (c.4) "hard masked feature modeling": similar to (c.2) but with zero vectors as mask. Formal definitions are detailed in Appendix B.6.

The results are depicted in Fig. 5c. In the figure, all paradigms outperform the discriminative approach except (c.2), which we attribute to the superior feature distribution modeling ability of generative paradigms. For (c.2), the semantic gap between different feature fields renders using a single mask vector for all fields inherently impractical. Therefore, adopting (c.3) with a field-aware mask significantly improves performance. Counterintuitively, a fixed zero vector outperforms learnable mask vectors. We hypothesize this discrepancy stems from differences between unsupervised and supervised generative paradigms. Using a learnable mask vector with supervised signals may impede feature distribution learning. Our "Predict All" paradigm

outperforms all others, demonstrating its superiority.

## 5. Related Works

**Feature-interaction-based recommender systems.** Designing improved feature interaction models has consistently represented a significant area of research within the field of recommender systems (Zhang et al., 2019; Cheng & Xue, 2021). A key focus in the advancement of modern recommendation systems is the development of more sophisticated feature interaction modules, including first-order (Richardson et al., 2007), second-order (Rendle, 2010; Pan et al., 2018; Sun et al., 2021), and high-order interactions (Lian et al., 2018; Wang et al., 2021; Li et al., 2024a). With the rise of deep learning, Deep Neural Networks (DNNs) with non-linear activation functions have been integrated into recommendation systems to capture implicit high-order feature interactions (Cheng et al., 2016; Guo et al., 2017; He & Chua, 2017; Lian et al., 2018; Wang et al., 2021). In addition to incorporating non-linearity in DNNs, several studies have explored the introduction of non-linearity in embeddings through gating mechanisms, such as FiBiNET (Huang et al., 2019), FinalMLP (Mao et al., 2023), and PEPNet (Chang et al., 2023). Orthogonal to these works, we propose a novel *Supervised Feature Generation* framework for CTR models, shifting from a discriminative "feature interaction" paradigm to a generative "feature generation" paradigm.

## 6. Conclusion

In conclusion, this work introduced a novel *Supervised Feature Generation* framework that shifts CTR modeling from discriminative feature interaction to generative feature generation. The framework's versatility was demonstrated through reformulating various existing feature interaction models into generative ones, ranging from explicit interaction models to complex DNN-based models. It could produce feature embeddings with reduced collapse and redundancy compared to raw ID embeddings.

## Acknowledgements

This work was supported by the National Natural Science Foundation of China (62441239, 62472394, U23A20319, 62441227, 62202443) as well as the Anhui Province Science and Technology Innovation Project (202423k09020011).

## Impact Statement

This paper presents work whose goal is to advance the field of Machine Learning. The research outcomes may have various societal implications, such as optimizing online advertising strategies or enhancing e-commerce functionalities, none of which we feel must be specifically highlighted here.

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

## A. Formal reformulation of existing feature interaction models

In Tab. 2, we provide the formal definition of how to reformulate existing discriminative models into generative paradigms. Notably, we only present the "feature interaction" or "feature generation" part in each paradigm for simplicity.

*Table 2.* Feature interaction models in discriminative & generative paradigm

| Model | Discriminative | Generative |
|---|---|---|
| FM | $\sum_{i=1}^{N}\sum_{j=i+1}^{N} \boldsymbol{v}_j \odot \boldsymbol{v}_i$ | $\sum_{i=1}^{N}\sum_{j=i+1}^{N} \boldsymbol{\sigma}([\boldsymbol{v}] \cdot W_{F(j)}) \odot \boldsymbol{v}_i$ |
| FmFM | $\sum_{i=1}^{N}\sum_{j=i+1}^{N} \boldsymbol{v}_j \odot [\boldsymbol{v}_i \cdot M_{F(i)\to F(j)}]$ | $\sum_{i=1}^{N}\sum_{j=i+1}^{N} \boldsymbol{\sigma}([\boldsymbol{v}] \cdot W_{F(j)}) \odot [\boldsymbol{v}_i \cdot M_{F(i)\to F(j)}]$ |
| CrossNet V2 | $\sum_{l=1}^{L}\sum_{i,j=1}^{N} \boldsymbol{v}_j^0 \odot (\boldsymbol{v}_i^l \cdot M_{F(i)\to F(j)}^l)$ | $\sum_{l=1}^{L}\sum_{i,j=1}^{N} \boldsymbol{\sigma}([\boldsymbol{v}]^l \cdot W_{F(j)}^l) \odot (\boldsymbol{v}_i^l \cdot M_{F(i)\to F(j)}^l)$ |
| DeepFM | $\sum_{i=1}^{N}\sum_{j=i+1}^{N} \boldsymbol{v}_j \odot \boldsymbol{v}_i + \text{DNN}([\boldsymbol{v}])$ | $\sum_{i=1}^{N}\sum_{j=i+1}^{N} \boldsymbol{\sigma}([\boldsymbol{v}] \cdot W_{F(j)}) \odot \boldsymbol{v}_i + \text{DNN}([\boldsymbol{v}])$ |
| xDeepFM | $\sum_{l=1}^{L}\sum_{i,j=1}^{N} \text{Conv}^l(\boldsymbol{v}_j^0 \odot \boldsymbol{v}_i^l) + \text{DNN}([\boldsymbol{v}])$ | $\sum_{l=1}^{L}\sum_{i,j=1}^{N} \text{Conv}^l(\boldsymbol{\sigma}([\boldsymbol{v}]^l \cdot W_{F(j)}^l) \odot \boldsymbol{v}_i^l) + \text{DNN}([\boldsymbol{v}])$ |
| IPNN | $\text{DNN}([[\boldsymbol{v}], \sum_{i=1}^{N}\sum_{j=i+1}^{N} \boldsymbol{v}_j \odot \boldsymbol{v}_i])$ | $\text{DNN}([[\boldsymbol{v}], \sum_{i=1}^{N}\sum_{j=i+1}^{N} \boldsymbol{\sigma}([\boldsymbol{v}] \cdot W_{F(j)}) \odot \boldsymbol{v}_i])$ |
| DCN V2 | $\sum_{l=1}^{L}\sum_{i,j=1}^{N} \boldsymbol{v}_j^0 \odot (\boldsymbol{v}_i^l \cdot M_{F(i)\to F(j)}^l) + \text{DNN}([\boldsymbol{v}])$ | $\sum_{l=1}^{L}\sum_{i,j=1}^{N} \boldsymbol{\sigma}([\boldsymbol{v}]^l \cdot W_{F(j)}^l) \odot (\boldsymbol{v}_i^l \cdot M_{F(i)\to F(j)}^l) + \text{DNN}([\boldsymbol{v}])$ |

## B. Detailed experimental configuration

### B.1. Dataset statistics

We adopt the Criteo x1 and Avazu x4 datasets provided by FuxiCTR (Zhu et al., 2020; 2022), whose statistics are summarized in Tab. 3.

### B.2. Implementation details of baseline methods

We first introduce common settings for all models: (1) For Criteo dataset, the embedding size is set to 10, batch size is set to 4,096, and learning rate is set to 1e-3. (2) For Avazu dataset, the embedding size is set to 16, batch size is set to 10,000, and learning rate is set to 1e-3. All experiments will be early stopped when results on validation dataset decrease for consecutive two training epochs.

Then we list the detailed setting of different baseline models. Notably, we do not tune these hyper-parameters when fitting these models into the proposed generative paradigm:

- FM: embedding regularization coefficient is set to 5.0e-06 for Criteo and 1.0e-06 for Avazu.

- FmFM: parameter regularization coefficients are set to 1.0e-06 for the both datasets; we adopt matrixed field embedding transform type (Sun et al., 2021) for both datasets.

- CrossNet V2: embedding regularization coefficient is set to 1.0e-05 and 0 for Criteo and Avazu, respectively; number of cross layers is set to 3, 5 for Criteo and Avazu.

- DeepFM: embedding regularization coefficient is set to 1.0e-05 and 0 for Criteo and Avazu, respectively; a parallel DNN with size [400, 400, 400] and [2000, 2000, 2000, 2000] are used for Criteo and Avazu, respectively.

*Table 3.* The number of user-item interactions of the adopted two datasets.

|  | Train | Valid | Test |
|---|---|---|---|
| Criteo | 33M | 8M | 4M |
| Avazu | 32M | 4M | 4M |

- xDeepFM: embedding regularization coefficient is set to 1.0e-05 and 0 for Criteo and Avazu, respectively; CIN hidden units are set to [16, 16] and [276] for Criteo and Avazu, respectively; DNN size is set to [400, 400, 400] and [500, 500, 500] for Criteo and Avazu, respectively.

- IPNN: embedding regularization coefficient is set to 1.0e-05 and 1.0e-09 for Criteo and Avazu, respectively; DNN size is set to [400, 400, 400] and [1000, 1000, 1000] for Criteo and Avazu, respectively.

- DCN V2: based on the setting of CrossNet V2, a parallel DNN with size [500, 500, 500] and [2000, 2000, 2000, 2000] are used for Criteo and Avazu, respectively.

All experiments can fit into a GPU with 14GB memories.

### B.3. Variation coefficients comparison between discriminative and generative paradigms

In Fig. 6, we have depicted the variation coefficients of the discriminative (DIS) and generative (GEN) paradigm concerning different datasets and recommendation performance metrics.

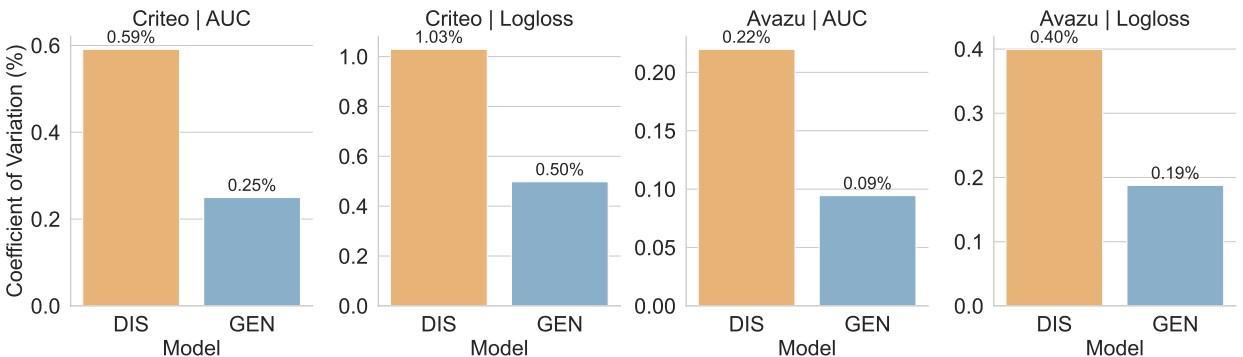

*Figure 6.* Variation coefficients comparison between discriminative and generative paradigms.

### B.4. Computational complexity analysis

Assuming the original model in the discriminative paradigm has complexity $O(A)$, the primary computational overhead when transitioning to a generative paradigm arises from the *encoder*. The encoder is implemented as a field-wise non-linear MLP, formally defined as:

$$f^i_{\text{encoder}}([\boldsymbol{v}]) = \sigma([\boldsymbol{v}]W_{F(i)}), \tag{9}$$

where $[\boldsymbol{v}] \in \mathbb{R}^{NK}$ denotes the concatenation of all feature embeddings, and $W_{F(i)} \in \mathbb{R}^{NK \times K}$ represents a field-wise weight matrix. Consequently, the total encoder complexity becomes $O(BLN^2d^2)$, where $B$ is the batch size, $L$ denotes the number of encoder layers, $N$ the number of feature fields, and $d$ the embedding dimension. This computational complexity aligns with mainstream discriminative feature interaction models (e.g., DCN V2), indicating comparable efficiency. Furthermore, as demonstrated in the *source* ablation study (Sec. 4.4), the complexity can be reduced to $O(BLN'^2d^2)$ by using all field embeddings as *source* only for fields with the lowest cardinality, achieving this optimization with moderate performance trade-offs.

> *Result 7. The extra computational burden introduced by reformulating existing discriminative feature interaction paradigms to the generative feature generation paradigm is marginal.*

### B.5. Robustness analysis of the batch-wise setting in Sec. 4.3.1.

In Sec. 4.3.1, we have conducted embedding analyses in dimensional collapse with a batch-wise setting, which greatly accelerates the analysis process compared with that based on the full validation dataset. But this batch-wise setting may introduce randomness to the analysis results, so we further provided the analysis of different seeds in Fig. 7. In the figure,

*Table 4.* Computational complexity when reformulating a discriminative feature interaction model into a generative feature generation model.The proposed generative paradigm achieves significant recommendation performance improvements, as detailed in Section 4.2, while incurring only a marginal increase in computational overhead—averaging 3.14% more computation time and 1.45% additional GPU memory consumption.

| | Model | | Criteo | | Avazu | |
|---|---|---|---|---|---|---|
| | | | Speed (time/epoch) | GPU memory (MB) | Speed (time/epoch) | GPU memory (MB) |
| Explicit | FM | DIS | 8m40s | 1600 | 3m08s | 2554 |
| | | GEN | 8m58s | 1605 | 3m13s | 2626 |
| | FmFM | DIS | 9m38s | 4846 | 3m48s | 10148 |
| | | GEN | 9m45s | 4892 | 3m57s | 10180 |
| | CrossNetv2 | DIS | 5m01s | 1050 | 1m43s | 2100 |
| | | GEN | 5m09s | 1096 | 1m57s | 2190 |
| DNN-based | DeepFM | DIS | 8m23s | 2090 | 4m36s | 3622 |
| | | GEN | 8m33s | 2122 | 4m44s | 3676 |
| | xDeepFM | DIS | 8m15s | 1906 | 2m43s | 3268 |
| | | GEN | 8m24s | 1908 | 2m45s | 3322 |
| | IPNN | DIS | 6m37s | 1544 | 3m06s | 2592 |
| | | GEN | 6m41s | 1558 | 3m14s | 2614 |
| | DCNv2 | DIS | 5m31s | 1238 | 5m44s | 2982 |
| | | GEN | 5m58s | 1282 | 6m01s | 3070 |

the trend of embedding spectra is consistent across all seeds, demonstrating the robustness of our batch-wise analysis setting. Specifically, on both Avazu and Criteo, the spectrum curves of discriminative paradigms exhibit an abrupt singular decay from $1 \times 10^{-5}$ to $1 \times 10^{-15}$, a reduction of $10^{10}$ times. This indicates a severe dimensional collapse issue. But in our generative paradigm, the abrupt singular value decay has been greatly alleviated. This verifies that the generative paradigm substantially mitigates the embedding dimensional collapse issue, forming a more balanced embedding space.

### B.6. Formal definition of different *target* design

We provide a detailed formal definition of the different *target* designs mentioned in Sec. 4.4.

(c.1) "Predict-random-selected", which generates only randomly selected feature fields:

$$y = \sum_i \sum_{j \in \mathcal{F}_{random}} \boldsymbol{\sigma}([\boldsymbol{v}] \cdot W_{F(i)}) \odot \boldsymbol{v}_j, \tag{10}$$

where $\mathcal{F}_{random}$ is a set of fields randomly sampled from all fields.

(c.2) "Masked feature modeling": randomly masking some fields in *source* with a learnable mask vector and predicting them as *target*, akin to masked image modeling (He et al., 2022):

$$y = \sum_i \sum_{j \in (\mathcal{F}_{unmask} \cup \mathcal{F}_{mask})} \boldsymbol{\sigma}([\boldsymbol{v}]_{\text{not masked}} \cdot W_{F(i)}) \odot \boldsymbol{v}_{j,mask} \tag{11}$$

where $\mathcal{F}_{unmask} \cup \mathcal{F}_{mask} = \mathcal{F}$, $\boldsymbol{v}_{j,mask} = \boldsymbol{mask}$ if j in $\mathcal{F}_{mask}$ else $\boldsymbol{v}_j$.

(c.3) "Field-aware masked feature modeling": similar to (c.2) but using field-specific mask vectors:

$$y = \sum_i \sum_{j \in (\mathcal{F}_{unmask} \cup \mathcal{F}_{mask})} \boldsymbol{\sigma}([\boldsymbol{v}]_{\text{not masked}} \cdot W_{F(i)}) \odot \boldsymbol{v}_{j,mask} \tag{12}$$

where $\mathcal{F}_{unmask} \cup \mathcal{F}_{mask} = \mathcal{F}$, $\boldsymbol{v}_{j,mask} = \boldsymbol{mask}_j$ if j in $\mathcal{F}_{mask}$ else $\boldsymbol{v}_j$.

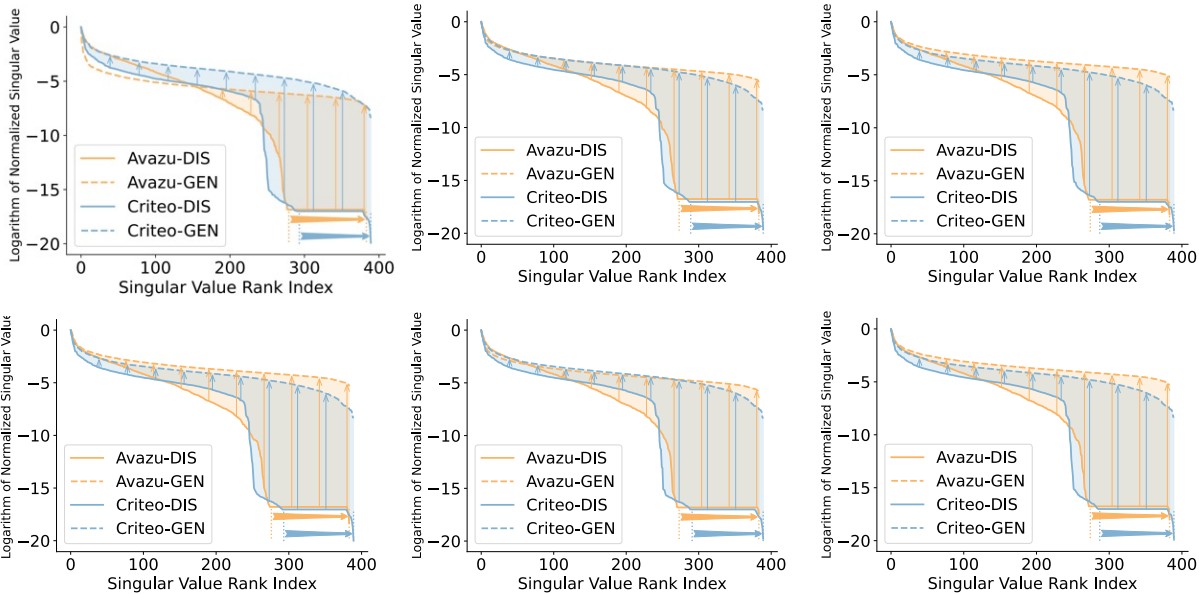

*Figure 7.* Normalized embedding spectrum visualization with batch-wise setting in different seeds. We can observe that the trend of embedding spectra is consistent across all seeds, which demonstrates the robustness of our batch-wise analysis setting.

(c.4) "Hard masked feature modeling": similar to (c.2) but with zero vectors as mask:

$$y = \sum_i \sum_{j \in (\mathcal{F}_{unmask} \cup \mathcal{F}_{mask})} \boldsymbol{\sigma}([\boldsymbol{v}]_{\text{not masked}} \cdot W_{F(i)}) \odot \boldsymbol{v}_{j,mask} \tag{13}$$

where $\mathcal{F}_{unmask} \cup \mathcal{F}_{mask} = \mathcal{F}$, $\boldsymbol{v}_{j,mask} = \boldsymbol{0}$ if j in $\mathcal{F}_{mask}$ else $\boldsymbol{v}_j$.

## C. Supplemental results

### C.1. Normalized singular value spectrum visualization of all models

The normalized singular value spectrum of all models are illustrated in Fig. 8. Similar to results concluded in Sec. 4.3.1, the feature generation framework substantially mitigates the embedding dimensional collapse issue, forming a more balanced and meaningful embedding space.

### C.2. Dimensional collapse analysis of embedding lookup tables

In Sec. 4.3.1, we focus on analyzing the spectrum of embeddings used to interact with the original embeddings, since we are mainly motivated to address the dimensional collapse issue of these embeddings. On the other hand, we can also follow Guo et al. (2024) to visualize the spectrum of embedding lookup tables, *i.e.*, $\boldsymbol{V}_i \in \mathbb{R}^{D_i \times K}$ defined in Sec. 2.2, where $i$ denotes one of the feature field, $D_i$ is the field's cardinality, and $K$ is the embedding dimension size of the embedding table. The results have been depicted in Fig. 9. In the figure, the spectrum of high-cardinality embedding lookup tables in the generative paradigm is higher than the discriminative one. This indicates the embedding space will be less dominated by some specific dimensions, which will greatly enhance the robustness of these embeddings. However, for those low-cardinality embeddings, the improvement remains limited. This is fundamentally because these field embeddings are inherently constrained by nature. For instance, the number of meaningful singular values of a matrix sized $4 \times K$ cannot exceed four.

### C.3. Pearson correlation matrix of all models

Similar to Sec. 4.3.2, we provide Pearson correlation matrix of all models on the Avazu dataset in Fig. 13. The conclusion remains the same as in Sec. 4.3.2: (1)There is a strong connection between redundancy reduction metric and recommendation performance: The most simple model FM yields the most matrix with intra-field and inter-field correlations, while the

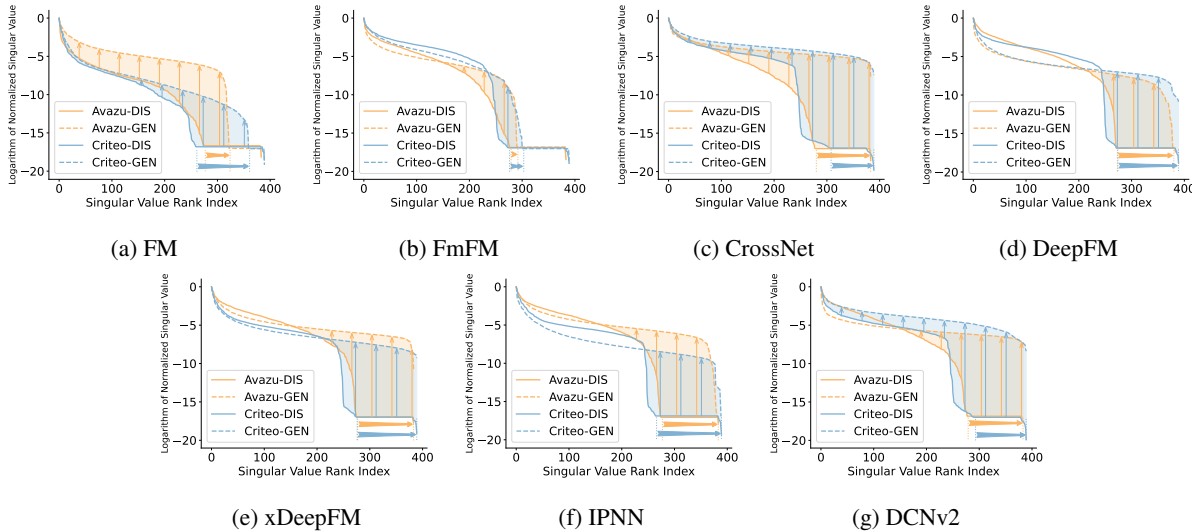

*Figure 8.* Normalized singular value spectrum of embeddings used to interact with raw ID embeddings. It is the concatenation of raw ID embeddings for the discriminative paradigm, while the embedding immediately constructed by the *encoder* for the generative paradigm.

correlation matrix of other models are reduced to some extent, depending on whether DNN (DeepFM, IPNN) or more advanced interaction modules (CrossNet V2, xDeepFM, DCN V2). (2)The feature generation framework produces embeddings highly de-correlated with raw ID embeddings: We can observe that the correlation matrices of all models become a nearly zero matrix within the generative paradigm.

### C.4. Comparison of different non-linear activation functions

We employ a field-wise non-linear single-layer MLP as our *encoder*, with the non-linear activation function being one of its most critical components. A natural question arises regarding the role of the non-linear activation function and the criteria for selecting an appropriate one. We have empirically assessed the effects of various activation functions on the *encoder*, with the findings illustrated in Fig. 10. As depicted in Fig. 10a, the absence of a non-linear activation function in the encoder results in a notable decline in performance, underscoring the importance of incorporating non-linearity within the *encoder*. Conversely, all non-linear activation functions enhance recommendation performance relative to the discriminative paradigm, with the rank of recommendation performance being *Sigmoid < Tanh < ReLU < SiLU*. Furthermore, we present the normalized singular value spectrum of embeddings in Fig. 10b. Initially, the spectrum of the linear activation is highly collapsed, potentially accounting for its inferior recommendation performance. Subsequently, it is observable that the spectra of all non-linear activation functions exhibit greater smoothness than that of the discriminative one. This suggests that non-linear activation functions play a pivotal role in alleviating the embedding dimensional collapse issue. Additionally, the spectrum adheres to the rank *Sigmoid < Tanh < ReLU < SiLU*, mirroring the ranking of recommendation performance. This observation further implies a strong correlation between the mitigation of embedding dimensional collapse and the enhancement of recommendation performance.

> *Result 8. The non-linear activation function is an important component of the field-wise MLP* encoder, *crucial for embedding dimensional collapse mitigation. Besides, many non-linear activation functions, including Sigmoid, ReLU, Tanh, and SiLU, can get consistent performance lift while mitigating the dimensional collapse.*

### C.5. Comparison with feature refinement and graph-based models

Some other methods also target enhancing the embeddings of CTR models with field graphs (Sun et al., 2022; Li et al., 2019; Wang et al., 2022c) or feature enhancement modules (Wang et al., 2023; 2022b;a). Our paradigm differs from these works in the sense that we aim to tackle the dimensional collapse issue due to the direct interaction of ID embeddings. We have empirically compared our paradigm with several representative feature refinement models, with results depicted in Tab. 5. We observed that some models outperform the discriminative DCNv2 models, but still underperform our generative

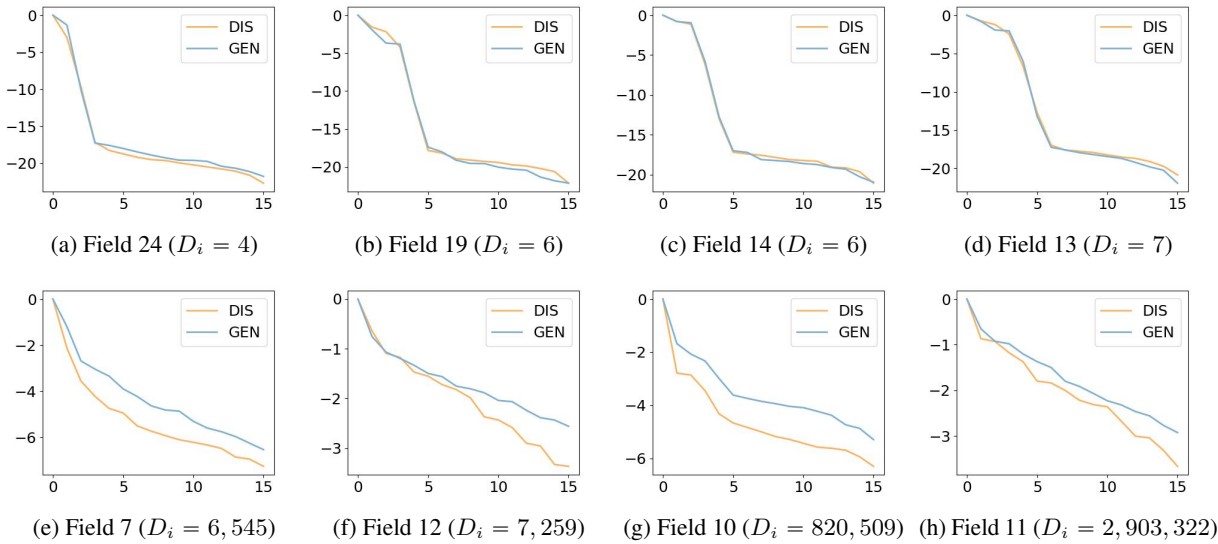

*Figure 9.* Normalized singular value spectrum of embeddings lookup tables $V_i \in \mathbb{R}^{D_i \times K}$, where $i$ denotes one of the feature field, $D_i$ is the field's cardinality, and $K$ is the embedding dimension size of the embedding table.

paradigm. Besides, we also studied the singular spectrum in Fig. 11, and we find that the feature enhancement methods can mitigate the dimensional collapse on the tail singular values compared to the vanilla discriminative DCN V2. However, our generative paradigm leads to more robust values across all dimensions.

*Table 5.* Comparison with other methods that also target enhancing embeddings of CTR models. We have compared with one classic field-graph method Fi-GNN (Li et al., 2019), and two feature enhancement methods GFRL (Wang et al., 2022a) and FRNet (Wang et al., 2022b). Notably, we also visualize the normalized spectrum of these methods in Fig.

| Model | | Criteo | | Avazu | |
|---|---|---|---|---|---|
| | | AUC↑ | Logloss↓ | AUC↑ | Logloss↓ |
| FiGNN | | 0.81352 | 0.43845 | 0.79156 | 0.37343 |
| DCNv2 | DIS | 0.81387 | 0.43826 | 0.79282 | 0.37222 |
| | GFRL | 0.81427 | 0.043773 | 0.79296 | 0.37194 |
| | FRNet | 0.81431 | 0.43789 | 0.79313 | 0.37191 |
| | GEN | **0.81472** | **0.43713** | **0.79342** | **0.37180** |

### C.6. T-SNE visualization comparison

In Fig. 12, we have visualized discriminative and generative embeddings with different cardinalities with T-SNE (Van der Maaten & Hinton, 2008). Fig. 12d and Fig. 12h depict embeddings of the highest cardinality field in the dataset, where we observe that the generative embeddings retain the separability as the discriminative paradigm. However, the improvement brought by the generative paradigm is substantial for embeddings of fields with less cardinality. In Fig. 12a, Fig. 12b, and Fig. 12c, the embeddings coalesce in the latent space, even for the field with the second-highest cardinality (Fig. 12c and Fig. 12g). After the generative reformulation, all three embeddings can form a more uniform distribution in the latent space, as illustrated respectively in Fig. 12e, Fig. 12f, and Fig. 12g. These results demonstrate that our generative paradigm can greatly improve the separability of embeddings, especially for embeddings with fewer cardinalities. This also supplements the aforementioned dimensional collapse phenomena analysis from a field-wise perspective.

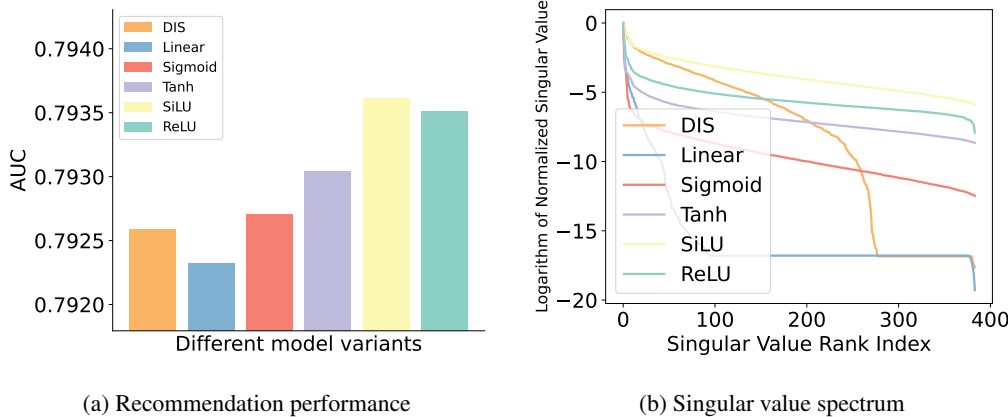

(a) Recommendation performance

(b) Singular value spectrum

*Figure 10.* We have implemented the *encoder* with different non-linear activation functions, including *ReLU*, *Sigmoid*, *Tanh*, and *SiLU*, and providing the corresponding results based on DCN V2: (a) The recommendation performance with different non-linear activation functions. (b) The normalized singular value spectrum of the embedding space with different non-linear activation functions.

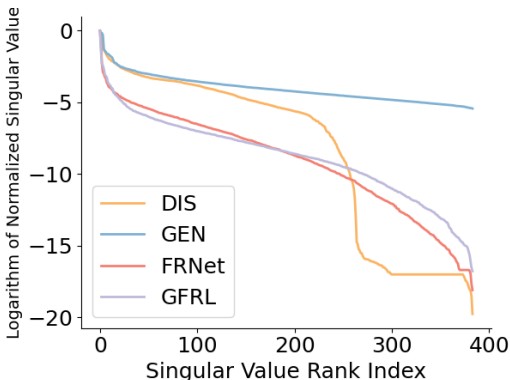

*Figure 11.* Normalized embedding spectrum of the feature enhancement methods. We can find that these feature enhancement methods can mitigate the dimensional collapse on the tail singular values compared to the vanilla discriminative DCN V2. However, our generative model leads to more robust values on all dimensions.

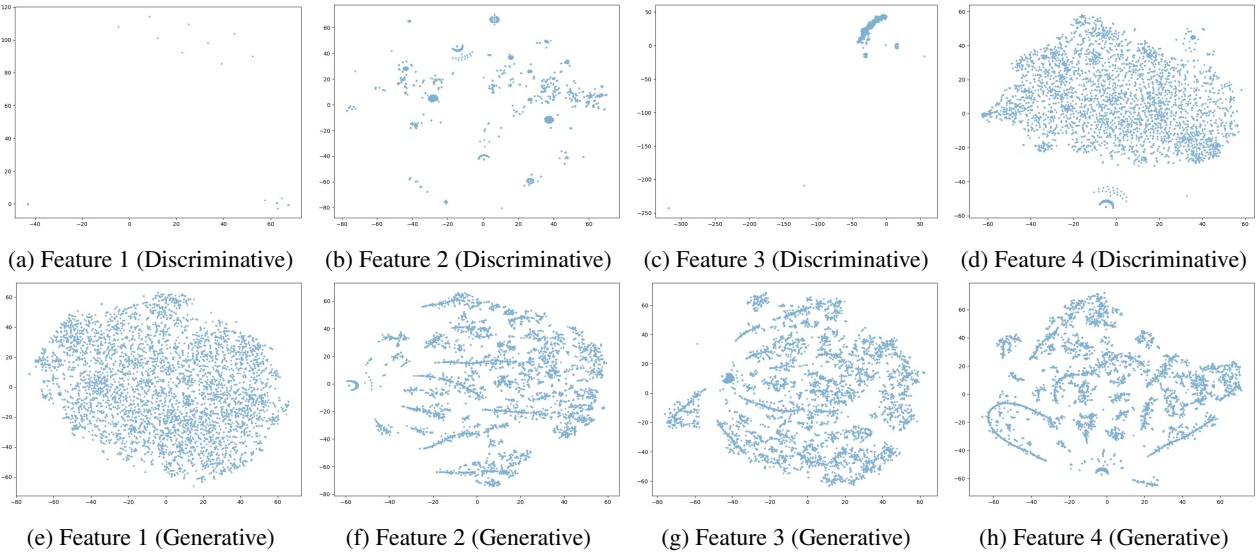

(a) Feature 1 (Discriminative)  (b) Feature 2 (Discriminative)  (c) Feature 3 (Discriminative)  (d) Feature 4 (Discriminative)

(e) Feature 1 (Generative)  (f) Feature 2 (Generative)  (g) Feature 3 (Generative)  (h) Feature 4 (Generative)

*Figure 12.* T-SNE visualisation of discriminative and generative embeddings of four features, numbered from 1 to 4. The cardinality of these features is 4, 4,051, 820,509, and 2,903,322, respectively. (a-d) illustrate embeddings of the four features within the discriminative paradigm; (e-h) illustrate embeddings of the four features within the generative paradigm.

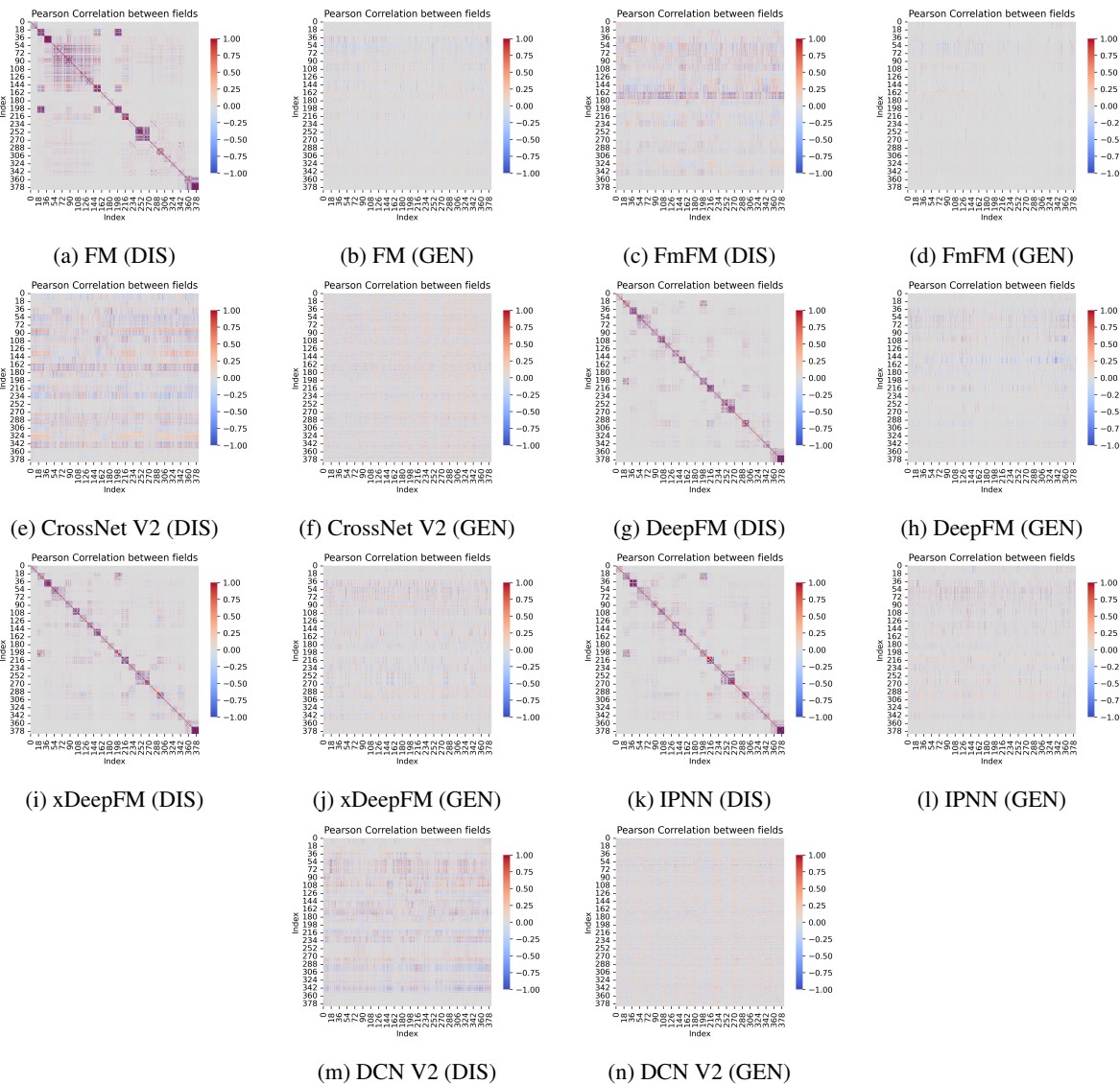

*Figure 13.* Pearson correlation matrix between two interacted embeddings. For all discriminative feature interaction models, the correlation matrix becomes a nearly zero matrix after reformulating them into a generative paradigm, which perfectly aligns with the redundancy reduction principle.

