# OpenReview forum: "From Feature Interaction to Feature Generation: A Generative Paradigm of CTR Prediction Models"
_ICML.cc/2025/Conference — ICML 2025 poster_

### Official Review · Reviewer_7DJA · 2025-03-07

**Overall Recommendation:** 3

**Summary:**

This paper highlights issues in discriminative CTR-based recommendation models, such as information redundancy and information collapse. To address these challenges, it proposes a feature generation framework that reformulates CTR prediction as a generative problem using a customized decoder network. The decoder network predicts all feature embeddings based on the input.

**Claims And Evidence:**

In Table 1, the authors demonstrate the consistent improvement of using the generative paradigm over the discriminative one, highlighting an increase in AUC and a reduction in log loss.

**Essential References Not Discussed:**

The key contribution is a paradigm shift from discriminative to generative models aimed at reducing information redundancy and mitigating dimensionality reduction. One important reference could be the inherently generative recommendation models with semantic IDs (Rajput et al. (2023)), comparing the results with the aforementioned references.

**Experimental Designs Or Analyses:**

In Table 1, can the authors shed some light on the hyperparameter setting for the decoder and MLP?

**Methods And Evaluation Criteria:**

The authors use two common evaluation metrics—Area Under the Curve (AUC) and log loss error reduction—to assess the model's quality. They also discuss the trade-off between quality and performance, noting the increase in computation time and memory usage.

**Other Comments Or Suggestions:**

no comments

**Other Strengths And Weaknesses:**

1. The paper is written with a clear flow.
2. The problem that the paper targets builds a bridge between the well-established discriminative recommendation models and the state-of-the-art generative models.
3. There is a lack of sufficient theoretical proof.
4. Some results seem slightly counterintuitive. Please refer to Q1.

**Questions For Authors:**

Q1. Can the authors elaborate on this observation? "On the other hand, increasing the complexity of the decoder with (b.3) significantly degrades recommendation performance, as the AUC decreases from 0.793512 to 0.792931. This may be caused by overfitting". Intuitively, a more complex decoder could capture more intricate relationships. What do the authors mean by overfitting?

Q2. Regarding result 5, do the authors have any theoretical or intuitive proof, aside from empirical evidence, to justify it?

Q3. Compared to inherently generative models in terms of AUC, log loss reduction is not included. If the generative recommendation model is indeed of higher quality, then this should be mentioned.

Q4. Can this approach lead to a paradigm shift for specific sequential recommendation models like Taobao Alibaba? Does the performance overhead, in this case, justify deploying this approach?

**Relation To Broader Scientific Literature:**

Although the authors focus on a specific problem within the recommendation models domain, the approach proposed in the paper can be applied to other domains to reduce information redundancy and dimensional collapse.

**Theoretical Claims:**

Given that the primary objective of this work is to reduce information redundancy and dimensional collapse, the theoretical justification is not adequately provided. Although the authors claim that both issues are mitigated empirically in lines 198-201, it remains unclear why passing raw ID embeddings through an MLP would effectively reduce information redundancy from a theoretical standpoint.

---

> ### Author Rebuttal · Authors · 2025-04-01
>
> We are grateful for your kind remarks. We hope the following responses can address your remaining concerns.
>
> ## Concerns about Theoretical Analysis
>
> > Response to "Theoretical Claims", and Point 3 of "Other Strengths And Weaknesses"
>
> Following DirectCLR[1], we have explored theoretical justification for our method's ability to mitigate collapse.
>
> Under gradient flow analysis (gradient descent with infinitesimal learning rate), the embedding update process follows a differential equation $\dot{v}_1 = - \frac{\nabla L}{\nabla v_1}$, where $v_1$ is a feature embedding and $L$ the loss. We aim to solve this equation and prove it will be rank deficient during training. **However, even for an FM with two features, the solution involves a complex Lambert W function[2]. We are still working on analyzing its rank properties.**
>
> [1] Understanding Dimensional Collapse in Contrastive Self-supervised Learning. ICLR 2022
>
> [2] On the Lambert W function. Advances in Computational mathematics, 1996.
>
> ## Questions about the Hyperparameter Setting
>
> > Response to "Experimental Designs Or Analyses"
>
> Our decoder is a one-layer MLP with fixed input and output dimensions. **The only tunable hyper-parameter is the nonlinear activations.** We evaluated different non-linear activations and found they are crucial for collapse mitigation (see Fig. 7 in Appendix D). Based on our experiments, we recommend ReLU or SiLU for the decoder.
>
> ## Discussion of Generative Sequential Recommendation with Semantic IDs
>
> > Response to "Essential References Not Discussed", and Q3 of "Questions For Authors".
>
> Thank you for your question. The suggested empirical comparison with them is difficult if not imposible since **Tiger is mainly designed for the sequential recommendaiton scenario while we focus on the feature interaction scenario**.
>
> ## Concerns about the Ablation Study
>
> > Response to Point 4 of "Other Strengths And Weaknesses", and Q1 of "Questions For Authors".
>
> Thank you for your insightful question. Regarding the overfitting issue, we acknowledge that our initial observation was speculative. To further investigate, we conducted a detailed spectral analysis comparing 1-layer and 2-layer MLPs, as shown in https://anonymous.4open.science/r/ICML2025-1748/supp/2layer.png.
>
> The figure shows that both 1- or 2-layer MLPs effectively alleviate the singular value decay (dimensional collapse) observed in discriminative paradigms. **However, singular values of the 2-layer MLP decline at a higher rate than the 1-layer MLP, suggesting that the extra layer leads to a more imbalanced embedding space.** This aligns with the recommendation performance.
>
> ## Theoretical or Intuitive Proof for Result 5
>
> > Response to Q2 of "Questions For Authors".
>
> Thanks for the question. We only have empirical evidence to justify result 5. We'll revise Result 5 as follows:
>
> "Result5. The field-wise non-linear one-layer MLP is a simple yet effective decoder. **Common** modifications, such as simplifying the model with field-shared MLPs or removing non-linearities, or increasing complexity through stacking MLP layers or self-attention, lead to inferior recommendation performance."
>
> We apologize for our imprecise statement and thank you for pointing this out, which helps us make the statement more rigorous.
>
> ## Question on Paradigm Shift for Sequential Recommendation
>
> > Response to Q4 of "Questions For Authors".
>
> **Many existing sequential recommendals already employ a generative paradigm under the next-item prediction framework**. Specially, pre-ranking models such as SASRec follows a self-supervised next-item generative paradigm, while ranking models such as DIN, DIEN follow a supervised next-item generative paradigm. The overhead of generative sequential models such as SASRec and DIN is low for short sequence, leading to a widely deployment of them in industrial systems. The computation cost becomes a challenge for long sequences, but there are many works (such as SIM, TWIN) to employ a two-stage Search & Modeling approach to resolve it.

---

### Official Review · Reviewer_z1Cm · 2025-03-12

**Overall Recommendation:** 3

**Summary:**

The paper “From Feature Interaction to Feature Generation: A Generative Paradigm of CTR Prediction Models” proposes a novel Supervised Feature Generation framework for Click-Through Rate (CTR) prediction models. The main algorithmic idea is to shift from the discriminative “feature interaction” paradigm to a generative “feature generation” paradigm. Instead of relying on raw ID embedding interactions, the framework predicts each feature embedding based on the concatenation of all feature embeddings.

The main findings indicate that the existing discriminative paradigm has limitations such as embedding dimensional collapse and information redundancy. The proposed generative paradigm mitigates these issues. Experimental results show that the framework can reformulate nearly every existing CTR model and brings significant performance improvements. Across different models, it achieves an average of 0.272% AUC lift and 0.435% Logloss reduction. It also reduces the embedding dimensional collapse and information redundancy, and has been successfully deployed in a large - scale advertising platform, leading to a 2.68% GMV lift in a primary scenario.

**Claims And Evidence:**

The claims are, in general, backed by various experiments, including the claims on embedding dimension collapse and redundancy reduction.

**Essential References Not Discussed:**

The paper does not have miseed reference as far as I know.

**Experimental Designs Or Analyses:**

The experiment design is sound, and I think the experiments back the main proposed claims well.

**Methods And Evaluation Criteria:**

The evaluation makes sense in general, where the mainstream recommendations algorithms are compared as baselines. Some details of the datasets are missing, and I hope the authors can add them.

**Other Comments Or Suggestions:**

This paper could be valuable for the industry recommendations. I think that many engineers and researchers in the industry may also be exploring this direction, and the results of thispaper  could be quite inspiring for industry recommendations.

**Other Strengths And Weaknesses:**

Advantage:
- The paper introduces an important research topic, especially for industry recommendations. The paper clearly states the disadvantage of discriminative models.
-  Experiments back up the claims that generative recommendations have an advantage over discriminative models. Moreover, the paper gives A/B test results. Though some details are missing (probably because of privacy or the double-blind policy), it makes the method much more convincing.


Disadvantage:
- The paper does not fully give every detail of methods and datasets, like the meaning of some notations in equation 2 and the dataset details in appendix B.1, including the dataset introduction, the number of users, items,etc. (If I missed the notations, please remind me). Particularly, without the explanation of important notations, It may get harder to understand the equation.

**Questions For Authors:**

- In Table 3 in the appendix, what is the meaning of the numbers in the dataset? Does it mean number of user-item interactions? how is the scale of users or items?
- In equation 2, what is the meaning of $l$ and $L$? What is meaning of $F(i),  F(j)$？

**Relation To Broader Scientific Literature:**

The paper is broadly related to the generative recommendation, which is the application of generative models for recommendation systems.

**Theoretical Claims:**

The paper does not contain theoretical claims.

---

> ### Author Rebuttal · Authors · 2025-04-01
>
> We are truly thankful for your review efforts. We apologize for the missing information on datasets and notations, and would clarify them as follows.
>
> ## Dataset Details
>
> > Response to "Methods And Evaluation Criteria", "Other Strengths And Weaknesses", and Q1 of "Questions For Authors"
>
> Thank you for your question. The numbers in the table are the number of samples, i.e., user-item interactions.
>
> As for the user or item scales, the user and item features are not annotated in the original dataset. But we can make the following guess:
> - [Criteo](https://github.com/reczoo/Datasets/tree/main/Criteo/Criteo_x1) has 13 numerical feature fields and 26 categorical feature fields. All 26 categorical features have been anonymized, but we can guess based on this assumption: user or item ID features are usually with the highest cardinalities. Feature fields with the top-5 cardinalities are (C3: 413,424), (C12: 409,749), (C21: 397,981) (C16: 365,811), (C4: 248,543), where the first denotes feature name and the second denotes the feature cardinality. These features have a relatively high probability of being user or item ID features. **Therefore, the scale of users and items could be on the order of ~100K**.
>
> - [Avazu](https://github.com/reczoo/Datasets/tree/main/Avazu/Avazu_x4) has 24 categorical features after preprocessing, part of which have been anonymized. Feature fields with the top-5 cardinalities are (device_ip: 2,903,322), (device_id: 820,509), (device_model: 7,259), (app_id: 6,545) (site_domain: 5,461). We infer that 'device' features represent users' devices, which are highly correlated with the number of users. **Therefore, the scale of users could be on the order of ~1M**. The remaining named features are not likely to be item features, so the item feature may be one of the anonymized features: (C14: 2,556), (C17: 434), (C20: 173). **Therefore, the scale of items could be on the order of ~1K**.
>
> Besides, we provide more statistics about our industrial dataset: **Our industrial model is trained on billions of samples daily, with hundreds of millions of unique users and around 1 million items**.
>
> ## Notations in Eq. 2
>
> > Response to "Other Strengths And Weaknesses", and Q2 of "Questions For Authors"
>
> Thank you for your question, and we apologize for missing explanations of Eq. 2. We clarify Eq. 2 as follows:
>
> - Eq. 2 is the formulation of DCNv2, which is a high-order feature interaction model. $L$ denotes the number of cross layers, $l$ denotes the layer index, $N$ denotes the total number of features, $i$ and $j$ denote the indices of features, $\mathbf{v}_i^{(0)} $ denotes the embedding of feature $i$ in the embedding layer, $\mathbf{v}_j^{(l)}$ denotes the embedding of the $j$-th term in the $l$-th layer.
> - $M_{F(i) \to F(j)}^{(l)}$ denotes the projection matrix between the $F(i)$ and $F(j)$ field pair in the $l$-th layer.
> - $F(i)$ and $F(j)$ denotes the field of feature $i$ and $j$, respectively.
> In addition, we will further review the entire manuscript to revise any potentially unclear sections and enhance its overall clarity.

---

> > ### Comment · Reviewer_z1Cm · 2025-04-03
> >
> > Thanks for the explanations, which addressed most of my concerns. I would raise the score to 3.

---

> > > ### Author Response · Authors · 2025-04-04
> > >
> > > We extend our sincere gratitude to Reviewer z1Cm for conducting a thoughtful reassessment of our work and for elevating the evaluation score. Your valuable feedback has prompted us to provide supplementary details regarding crucial experimental aspects and clarify key manuscript notations. We are deeply encouraged by the successful resolution of the issues you raised.

---

### Official Review · Reviewer_SGPG · 2025-03-13

**Overall Recommendation:** 4

**Summary:**

This paper introduces a feature generation framework that reformulates conventional CTR models through a generative paradigm, effectively addressing dimensional collapse and information redundancy issues in feature embeddings. The claims are substantiated by rigorous empirical evidence spanning widely adopted benchmark datasets. The proposed methodology is thoroughly evaluated under well-designed experimental settings, demonstrating consistent superiority over baseline approaches. The core implementation is publicly accessible in the supplementary materials to ensure reproducibility. While a related work exploring embedding collapse phenomena is cited, its methodological distinctions from the current approach warrant deeper analysis.

Overall, this paper studies a foundational problem in CTR prediction by innovatively bridging generative paradigms with feature interaction models, with well-designed experiments and insightful conclusions to support the proposed method. I am inclined to recommend acceptance.

**Claims And Evidence:**

This paper claims to reformulate existing feature-interaction models into a novel feature generation paradigm. This claim is substantiated through comprehensive comparative experiments and ablation studies.

It claims to mitigate the inherent drawbacks of conventional ID embeddings in traditional feature interaction models, i.e., dimensional collapse and information redundancy. This claim is validated via two inspiring and sound experiments.

**Essential References Not Discussed:**

Although mentioned, the multi-embedding[1] method is not fully discussed in the paper, which is also specially designed for embedding dimensional collapse mitigation.

[1] On the Embedding Collapse when Scaling up Recommendation Models. International Conference on Machine Learning. PMLR, 2024.

**Experimental Designs Or Analyses:**

I have verified the experimental validity, including the main comparison, embedding analysis, and the ablation studies.

Strengths:

- The proposed paradigm delivers substantial performance improvements across diverse existing CTR models, with successful deployment in production-scale advertising systems.
- The analysis experiments are well-designed, validating that the proposed method can address the claimed dimensional collapse and information redundancy issues. The correlation shift trend from weak to strong models is insightful.
- The authors conduct a systematic investigation of paradigm components through well-structured ablation studies.

Weaknesses:

- The analysis experiments employ a batch-wise processing. I'm concerned the results may be inconsistent on the full validation dataset.

**Methods And Evaluation Criteria:**

The proposed method establishes a feature generation paradigm designed to address feature embedding challenges. The framework's effectiveness is validated through empirical evaluation, employing well-established baseline models and standardized datasets consistent with common protocols.

**Other Comments Or Suggestions:**

Typos
	- "hadamard product" should be "Hadamard product"
	- Line 113, "has focus" -> "has focused"
	- An extra annotated "(1)" in Line 269

**Other Strengths And Weaknesses:**

Overall, this is an interesting paper that studies a fundamental problem in a real-world application problem. PLS see the "Experimental Designs Or Analyses" section.

**Questions For Authors:**

Will the analysis results be inconsistent when adopted on the entire validation dataset?
What is the relationship between this work and the multi-embedding[1] method in embedding collapse mitigation?

[1] On the Embedding Collapse when Scaling up Recommendation Models. International Conference on Machine Learning_. PMLR, 2024.

**Relation To Broader Scientific Literature:**

The dimensional collapse issue in feature embeddings has been investigated in recent literature[1], where this challenge was effectively addressed via a multi-embedding method.

[1] On the Embedding Collapse when Scaling up Recommendation Models. International Conference on Machine Learning_. PMLR, 2024.

**Theoretical Claims:**

This paper does not make theoretical claims.

---

> ### Author Rebuttal · Authors · 2025-04-01
>
> We sincerely thank you for your valuable comments. We hope the following responses can address your concerns.
>
> ## Concerns about Batch-wise Analysis
>
> > Response to Weaknesses of "Experimental Designs Or Analyses" & Q1 of "Questions For Authors".
>
> Thank you for your suggestion. The suggested analysis of the entire validation dataset is time-consuming, so we have adopted this batch-wise setting. To ensure the experiment's robustness, we have **repeated the analysis experiments with 6 different random seeds** when sampling the batches, with results in https://anonymous.4open.science/r/ICML2025-1748/supp/seed.jpg. **The trend of embedding spectra is consistent in all batches**: On both Avazu and Criteo, the spectrum curve of discriminative paradigms exhibit an abrupt singular decay from ~$1\times 10^{-5}$ to ~$1\times 10^{-15}$, a reduction of $10^{10}$ times. This indicates a severe dimensional collapse issue. But in our generative paradigm, the abrupt singular value decay has been greatly alleviated. This verifies that the generative paradigm substantially mitigates the embedding dimensional collapse issue, forming a more balanced embedding space.
>
> ## Relationship to Multi-Embedding
>
> > Response to "Essential References Not Discussed" & Q2 of "Questions For Authors".
>
> Thanks for your question. **The proposed paradigm is orthogonal to the multi-embedding method, and these two methods can be seamlessly combined.** We have conducted experiments based on one of the most representative models DCNv2 on the Avazu dataset, with results presented as follows:
>
> - Recommendation performance (AUC) comparison.
>
>   | Model\Embedding size | 16          | 16 $\times$ 2 | 16 $\times$ 4 | 16 $\times$ 8 | 16 $\times$ 10 |
>   |----------------------|-------------|---------------|---------------|---------------|----------------|
>   | DCNv2 - DIS          | 0.79282     | 0.79402       | 0.79434       | 0.79539       | 0.79577        |
>   | DCNv2 - GEN          | **0.79342** | **0.79469**   | **0.79534**   | **0.79599**   | **0.79617**    |
>
>   We can observe that, the generative DCNv2 with multi-embedding outperforms discriminative DCNv2 with multi-embedding.
>
> - Embedding spectrum analysis. We have illustrated the spectrum of DCNv2 (DIS), DCNv2 (GEN), and DCNv2(GEN + MultiEmbedding) in https://anonymous.4open.science/r/ICML2025-1748/supp/collapse.png. The results show that:
>
>   - **Embedding collapse of DCNv2 (DIS)**: The spectrum curve of DCNv2 (DIS) exhibits a dramatic decay after the singular value index 250, which indicates a collapsed embedding space.
>   - **Embedding robustness of DCNv2 (GEN)**: Different from DCNv2 (DIS), the spectrum curve of DCNv2 (GEN) does not exhibit the abrupt decay. Instead, they decline slowly, indicating a more balanced embedding space.
>   - **Multi-embedding alleviates collapse**: DCNv2(GEN + MultiEmbedding) can lead to a slightly slower decline rate than DCNv2 (GEN), which indicates a more robust embedding space.
>
> We compute the Information Abundance[1] (IA) values of these 3 variants respectively: 9.0082, 13.5031, 13.6999. A higher IA value indicates a less-collapsed embedding space. The IA results are consistent with the embedding spectrum figure. **These spectra and IA results both demonstrate that our method and multi-embedding can be effectively combined to achieve both better performance and less collapse.**
>
> [1] Guo, Xingzhuo, et al. On the Embedding Collapse when Scaling up Recommendation Models. ICML, 2024.
>
>
> ## Typos
>
> > Response to "Other Comments Or Suggestions"
>
> Thank you for your corrections and we will modify the corresponding symbols accordingly. In addition, we will further review the entire manuscript to revise any potentially unclear sections and enhance the overall clarity of our work.

---

> > ### Comment · Reviewer_SGPG · 2025-04-04
> >
> > I appreciate the authors' responses, which have effectively resolved my initial concerns:
> > - The experimental validation demonstrates consistent performance patterns across multiple data batches, reinforcing the methodological robustness.
> > - By building upon established multi-embedding frameworks, the proposed approach achieves significant performance gains. The extra spectral analysis further suggests the potential of combining these two methods for mitigating the embedding dimensional collapse issue.
> > I respect the authors for their rigorous experiments and recommend incorporating these findings into the final manuscript to strengthen its arguments.
> >
> > Having carefully reviewed the other comments, I maintain my positive assessment. This work makes a valuable contribution by shifting CTR models from discriminative to generative paradigms, effectively mitigating persistent challenges like embedding collapse and information redundancy. The comprehensive analysis experiments offer insights that could guide the design of future CTR models.

---

> > > ### Author Response · Authors · 2025-04-05
> > >
> > > We sincerely appreciate Reviewer SGPG for re-evaluating our paper and raising the score. We have carefully incorporated your valuable suggestions, especially regarding the analysis experiment robustness and multi-embedding discussion, and have made thorough efforts to address all concerns through additional experiments and detailed explanations. Your insightful feedback has significantly enhanced our work, and we are grateful that we were able to address your concerns satisfactorily.

---

### Official Review · Reviewer_Rab2 · 2025-03-22

**Overall Recommendation:** 2

**Summary:**

This paper proposes a Supervised Feature Generation (SFG) framework that reformulates the conventional discriminative CTR prediction paradigm into a generative paradigm. Rather than modeling direct interactions among raw ID embeddings, the proposed method generates each feature embedding based on the concatenation of all other feature embeddings. The goal is to address two issues in CTR modeling: embedding dimensional collapse and information redundancy. The framework is designed to be model-agnostic and can be applied to many standard CTR models such as FM, DeepFM, CrossNet, and DCN V2. Experiments on public benchmarks and an online A/B test on a large-scale advertising platform show small improvements in AUC and Logloss, as well as practical gains in industrial deployment scenarios.

**Claims And Evidence:**

The paper makes several claims, including:

- That the proposed generative paradigm leads to more semantically meaningful feature embeddings.

While the framework is novel in formulation, the motivation behind the paradigm shift is weakly supported. The core claim that “we shift from raw ID embedding interactions to semantically meaningful feature generation” is misleading. In practice, most modern CTR models do not rely solely on raw ID embedding interactions; they use cross networks or deep modules to overcome known limitations. Therefore, the paper constructs a false dichotomy by framing the entire feature interaction paradigm as flawed, when in fact the issue lies with simplistic interaction mechanisms.

- That it provides a general-purpose enhancement applicable to a wide range of CTR models.

Furthermore, although performance improvements are reported (e.g., ~0.272% AUC lift), they are relatively modest considering the additional model complexity. The method increases computation time by 3.14% and GPU memory by 1.45%, raising questions about the cost-effectiveness of the generative design, especially for production systems where latency and efficiency are critical.

**Essential References Not Discussed:**

Please check recent embedding refinement, denoising, or feature interaction methods based on field graphs (there are many) for recommendation systems.

**Experimental Designs Or Analyses:**

The experiments are fine.

**Methods And Evaluation Criteria:**

The paper does not make it entirely clear why the embedding reconstruction process should yield better representations in all settings. The notion of “generating semantically meaningful features” remains vague, and the benefit of reconstruction versus learned interaction is not theoretically or empirically justified beyond intuitive arguments.

**Other Comments Or Suggestions:**

"Inherent drawbacks of raw ID embeddings. Embed-
dings of low-cardinality feature fields only span a low-
dimensional embedding space, intrinsically leading to a
bottleneck for representing abundant information. More-
over, according to the interaction collapse theory (Guo
et al., 2024), direct interactions between raw ID embed-
dings can lead to severe dimensional collapse issue (Jing
et al., 2021). Consequently, even embeddings of high-
cardinality feature fields will be constrained to a low-
dimensional subspace of the available embedding space,
thereby limiting their information abundance." This paragraph needs to be explained or corrected.

**Other Strengths And Weaknesses:**

Pros:

- Conceptually novel formulation with a modular implementation.

- Broad compatibility with many CTR models.

- Real-world deployment.

Cons:

- Why generative paradigm is needed is still unclear. Overstated framing of the paradigm shift—feature interaction is mischaracterized.

- Modest offline improvements raise concerns about cost-effectiveness. The computation cost is less mentioned.

- Vague definition of “semantic generation”; unclear benefit of reconstructing embeddings over learning them directly.

**Questions For Authors:**

Please check my comments above.

**Relation To Broader Scientific Literature:**

The paper relates to feature embedding learning, representation redundancy, and autoencoding principles. However, it does not clearly position itself against existing embedding refinement, denoising, or feature interaction methods based on field graphs (there are many), which may share similar goals with more principled frameworks.

**Theoretical Claims:**

There is no formal theoretical contribution.

---

> ### Author Rebuttal · Authors · 2025-04-01
>
> We thank the reviewer for their comments. We have addressed the comments in the rebuttal below.
>
> ## Clarification on the Motivation.
>
> > Response to Point 1 of "Claims And Evidence" & "Methods And Evaluation Criteria".
>
> We'll elaborate more on our claim. Our work is mainly inspired by the Interaction-Collapse Theory, that is, **direct interactions between ID embeddings in existing CTR models with cross networks lead to dimensional collapse**. We resolve this challenge by shifting from the discriminative paradigm that involves direct interactions between raw ID embeddings to a generative paradigm **that constructs new embeddings by a decoder network and interacts the constructed embeddings with the raw ID embeddings**.
>
> We don't aim to claim  the entire feature interaction paradigm as flawed, and **we apologize of we made such unintentional and misleading claim**. As researchers in this area, we appreciate the promising progresses in the last decade. In particular, we'd like to recognize the **positive impact of cross network and DNN** as follows, and will add them in the revised version.
>
> The cross network, especially CrossNet in DCN V2, has been proved effective in CTR prediction. A recent work [1] validates that its cross function mitigates dimensional collapse to some extent via field-pair-wise transformation matrix. However, we find that our generative variants can further improve the embeddings' dimensional robustness (Fig 3.c).
>
> Regarding deep networks, **DNNs can also mitigate dimensional collapse compared to direct cross networks**. In fact, **we have a submited paper studying DNNs in feature interaction models from this perspective**, showing that **non-linear activations greatly improve dimensional robustness**. We will add this discussion to the revised version.
>
> [1] Towards Unifying Feature Interaction Models for Click-Through Rate Prediction. 2024.
>
> ## Concerns about the Cost-effectiveness.
>
> > Response to Point 2 of "Claims And Evidence".
>
> We thank the reviewer for pointing out the cost-effectiveness trade-off. In industry CTR prediction, even a 0.1% AUC lift is considered significant [1]. Our analysis shows **the ROI (GMV lift/computation cost) for several scenarios far exceeds our release threshold** (typically in the dozens; we omit the exact value since it's a commercial secret). Since February, **three generative models have passed performance/cost reviews and been fully deployed**.
>
> [1] FuxiCTR: An Open Benchmark for Click-Through Rate Prediction. 2020.
>
> ## Discussion on Essential References.
>
> > Response to "Relation To Broader Scientific Literature" & "Essential References Not Discussed".
>
> Thanks for your valuable suggestion. We will specifically discuss these methods in the Related Works section of the final paper. We list the main differences as follows due to space limits:
>
> Our paradigm differs from these works in the sense that **we aim to tackle the dimensional collapse issue due to the direct interaction of ID embeddings**. We argue that the above-mentioned related works can't achieve this by refinement, denoising or adopting a GNN architecture.
>
> We empirically compared our paradigm with several representative feature refinement models, with results as follows. We observed that some models outperform the discriminative DCN V2 models, but still underperform our generative model.
>
> | Model |  | Criteo | Avazu |
> |:---:|:---:|:---:|:---:|
> | FiGNN[3] | - | 0.81352 | 0.79156 |
> | DCNv2 | DIS | 0.81387 | 0.79282 |
> | GFRL[1] | - | 0.81427 | 0.79296 |
> | FRNet[2] | - | 0.81431 | 0.79313 |
> | DCNv2 | GEN | 0.81472 | 0.79342 |
>
> We also studied the singular spectrum and found that **they can mitigate the dimensional collapse on the tail singular values**compared to the vanilla discriminative DCN V2. However, **our generative model leads to more robust values on all dimensions**. Refer to the spectrum analysis in https://anonymous.4open.science/r/ICML2025-1748/supp/refinement.png.
>
> [1] MCRF: Enhancing CTR Prediction Models via Multi-channel Feature Refinement Framework.
>
> [2] Enhancing CTR prediction with context-aware feature representation learning.
>
> [3] Fi-gnn: Modeling feature interactions via graph neural networks for ctr prediction.
>
> ## Clarification of the Discussion Parapragh.
>
> > Response to "Other Comments Or Suggestions:"
>
> We will elaborate on this paragraph and revise the manuscript accordingly with the following:
>
> **Dimensional Collapse of Raw ID Embedding Interaction**. The embeddings of some fields may only span a low-dimensional space due to various reasons, such as the low cardinality of this field. For example, the embeddings of the gender field with values of Male, Female, and Unknown can only span a 3-dimensional space. According to the Interaction-Collapse-Theory, **the interactions with these low-dimensional field embeddings may lead to the dimensional collapse of the embeddings of the other fields**.

---

### Decision · Program_Chairs · 2025-05-01

**Decision:**

Accept (poster)

**Comment:**

Most of the reviewers have favorable opinions; the first reviewer's negative comment can be fixed before the camera-ready deadline.